# The gate injection-based field-effect synapse transistor with linear conductance update for online training

Seokho Seo [1,2], Beomjin Kim[1,2], Donghoon Kim[1,2], Seungwoo Park [1,2], Tae Ryong Kim[1], Junkyu Park[1], Hakcheon Jeong [1], See-On Park [1], Taehoon Park [1], Hyeok Shin [1], Myung-Su Kim [1], Yang-Kyu Choi[1] & Shinhyun Choi [1] ✉

Neuromorphic computing, an alternative for von Neumann architecture, requires synapse devices where the data can be stored and computed in the same place. The three-terminal synapse device is attractive for neuromorphic computing due to its high stability and controllability. However, high non-linearity on weight update, low dynamic range, and incompatibility with conventional CMOS systems have been reported as obstacles for large-scale crossbar arrays. Here, we propose the CMOS compatible gate injection-based field-effect transistor employing thermionic emission to enhance the linear conductance update. The dependence of the linearity on the conduction mechanism is examined by inserting an interfacial layer in the gate stack. To demonstrate the conduction mechanism, the gate current measurement is conducted under varying temperatures. The device based on thermionic emission achieves superior synaptic characteristics, leading to high performance on the artificial neural network simulation as 93.17% on the MNIST dataset.

In the advent of the big data era, the dramatic advance of machine learning technology and artificial intelligence have occurred, demanding the computing ability to handle the data-intensive task[1]. However, the currently exploited conventional von Neumann architecture has become the bottleneck due to its limitation to parallel computing ability and high power consumption to deal with the big data, caused by obligated data transfer through the data bus between the physically separated processing unit and memory[2–4]. Therefore, to perform successful big data analysis, new computing architectures have been developed. The main key idea of the new architectures is to compute the data in memory without data transfer (or small movement of data), enabling reducing power consumption and suppressing latency by parallel data processing ability[4–7].

Neuromorphic computing is one of the candidates for post-von Neumann architecture. By mimicking the synaptic behavior of the biological neural network, the big data can be processed by parallel computing in an energy-efficient way in real-time[5,8,9]. For accelerating the artificial neural network (ANN) with this new architecture, the neuromorphic device, which can memorize and compute the data on the same device, is required. Recently, several studies utilizing conventional memory, such as DRAM and charge trap flash memory (CTF), and emerging memory devices such as PRAM and ReRAM have been reported in neuromorphic applications[10–17].

In the case of conventional memories, the well-established DRAM secures fast write speed and linear conductance update[10]. Capacitor-based synaptic devices with a DRAM-like structure also have main advantages in online training for repeated updates because of their high endurance[18]. However, because of poor retention characteristics, the weight values must be transferred to nonvolatile memories very frequently during the training process, resulting in high power

---

[1]The School of Electrical Engineering, Korea Advanced Institute of Science and Technology (KAIST), Daejeon 34141, Republic of Korea. [2]These authors contributed equally: Seokho Seo, Beomjin Kim, Donghoon Kim, Seungwoo Park. ✉e-mail: shinhyun@kaist.ac.kr

consumption. Moreover, these devices are difficult to create and retain analog conductance states with a single device and require a capacitor (storing charges for weight values) and several additional transistors to implement analog states[10,19,20]. This means that it has a drawback in terms of device integration density compared to a single synaptic device.

On the other hand, nonvolatile memories such as CTF can distinguish between states of multi-level cells depending on how many charges are trapped in the charge trap layer, and have long retention[12]. Additionally, several studies show that the endurance characteristics of CTF can be significantly improved by structural and material engineering of the device[21–23]. Therefore, research using CTF devices is being actively conducted for applications in neuromorphic computing, as well as for the main memory for data storage. However, while data can be stored for long periods without data loss, CTF normally has a large operation voltage and slow speed, requiring more energy, especially for data erasing[24–26]. In the case of online training, using devices with low update energy is advantageous because the training demands repeated writing and erasing operations more than millions of times.

The two-terminal emerging memories have been extensively studied as a promising candidate among neuromorphic devices due to their simple structure and scalability. Furthermore, they can be integrated into large-scale crossbar-array for vector-matrix multiplication, which is essential for the basic operation of neuromorphic computing[8,27–29]. However, their device variation caused by randomly formed filament during set/reset. This stochastic behavior of ion movement causes unreliable variation and it has been the significant bottleneck for the successful application as a computing device[4,28]. On the other hand, the three-terminal synaptic device has the advantage of enhancing synaptic weight controllability, and allows simultaneous reading and writing the data[8,9,30–32].

Besides suppressing the problems mentioned above, synapse device characteristics such as high linear conductance update and compatibility with the conventional Complementary-Metal-Oxide-Semiconductor (CMOS) system are commonly required to acquire the high performance of crossbar-array based neuromorphic computing as that of software-based ANN[32–36]. Especially, the linearity of the Long-Term Potentiation and Long-Term Depression (LTP-LTD) is regarded as one of the most important characteristics for synapse device evaluation[33]. By achieving the linear conductance update with identical consecutive pulse scheme, it is believed to enable the multi-level operation while reducing the burden on peripheral circuits to operate crossbar array[12,37].

The conventional three-terminal floating gate-based flash memory shows high nonlinearity in weight updates[12,30,38,39] due to the Fowler-Nordheim (F-N) tunnelling, a vital function of the electric field changed by electrons stored charge state[38,40]. Ion-conducting electrolyte-based three-terminal synapse devices show high linear conductance update for weight state[41–44]. However, they are vulnerable in the perspective of low on/off ratio, high programming pulse width, and incompatibility with conventional CMOS devices.

In this paper, we propose a three-terminal Gate Injection-based Field-Effect Transistor (GIFET), which utilizes the CMOS compatible material and fabrication process. Through different operation mechanisms from conventional flash memory, we derive superior synapse device characteristics, such as high linearity and symmetry, high temporal and spatial uniformity (<1.64%, 9.76%), and low power consumption (50 fJ/SOP). These performances lead to a high accuracy of approximately 93.17% with the MNIST handwritten recognition dataset.

## Results

### Structure and operation principle of GIFET

The dependence of current density through tunnelling oxide on the current floating gate charge state triggers the nonlinearity of conventional flash memory due to its primary update mechanism, F-N tunnelling[38–40]. To relieve the current density dependence on floating gate charge, we program and erase the charges in the stored layer by thermionic emission of an electron to and from the gate metal (see Fig. 1a), which is the weaker function of electric field than field emission[45,46].

As the current density through the thermionic emission depends on the barrier height between each layer, the band diagram of GIFET was designed as shown in Fig. 1b. The barrier height between the charge store layer (CSL) and blocking layer should be moderately low to guarantee sufficient current density for a high on/off ratio and to hold electrons, except during write/erase operation for high retention. Therefore, we selected a CSL material ($\chi_{CSL} = 4.92 eV$)[47] with a greater electron affinity than the blocking layer ($\chi_{BL} = 3.93 eV$)[48].

Figure 1c shows the cross-sectional Transmission Electronic Microscope (TEM) image of the device. As presented, 20 nm-thick n$^+$ doped Si layer was utilized on silicon dioxide layer to design device area by mesa pattern. On top of that, silicon dioxide (SiO$_2$) was applied as the gate oxide. WO$_x$ was deposited as CSL with high electron affinity to form a shallow well in the energy band diagram, where the electrons are stored. Subsequently, a-Si:H was utilized as a blocking layer for a lower barrier height difference with gate metal (see details in Methods and Fig. 1d for Second Ion Mass Spectroscopy (SIMS) data of the gate stack of the GIFET). The amount of charge stored in the WO$_x$ layer widens or shortens the Si channel's depletion region by field-effect, and this changing depletion region is the primary mechanism to control artificial synapse weight in terms of the conductance.

Figure 1e–g present the schematics of basic operations of the GIFET for write, erase, and read process, respectively. When a positive bias is applied on the gate while source and drain are grounded, negative charge electrons on WO$_x$ layer are extracted to gate metal due to electric field between gate and channel (see Fig. 1e). Therefore, the depletion region in the channel decreases, leading to increasing channel conductance (write process). Reversely, when a negative bias is applied on the gate, and channel source-drain are grounded, electrons are injected from gate metal to the WO$_x$ layer, decreasing channel conductance (erase process, see Fig. 1f). In order to read the stored weight of a device, the gate is grounded, and read voltage is applied to drain to confirm the current weight state of the memory cell by measuring the conductance of the cell (see Fig. 1g). More detailed operating principles are described in Supplementary Fig. 1. In addition, because GIFET is a transistor-based device, its electrical characteristics are measured and evaluated as a transistor (see Supplementary Fig. 2). As the number of electrons stored in the CSL by the write/erase operation varies, the $I_D$-$V_G$ and $I_D$-$V_D$ characteristics can be changed, which means that the weight of the synaptic device can be controlled.

### The relationship between linearity and conduction mechanism

To investigate the effect of the conduction mechanism for charge transport through the blocking layer on the linearity, we observed the dependence of the current density through the blocking layer at several temperatures. To focus on the current through the blocking layer and confirm the presence of the Schottky barrier, a gate stack of the device without the gate oxide was prepared as shown in Fig. 2a. I-V characteristic measurements were conducted in the temperature range 273 K–423 K.

Figure 2b presents the Arrhenius plot of $\ln(I/T^2)$ versus $q/kT$. The linear relationship between $\ln(I/T^2)$ and $q/kT$ is observed, which implies the existence of the Schottky barrier, and the primary conduction mechanism through the blocking layer is thermionic emission current[49]. More details on barrier height are in Supplementary Fig. 3. On the other hand, we also observed the current density of the gate stack with SiO$_x$ interfacial layer between CSL and blocking layer to see the temperature dependence (see Supplementary Fig. 4a). During the write operation, the SiO$_x$ interfacial layer is under a higher electric field

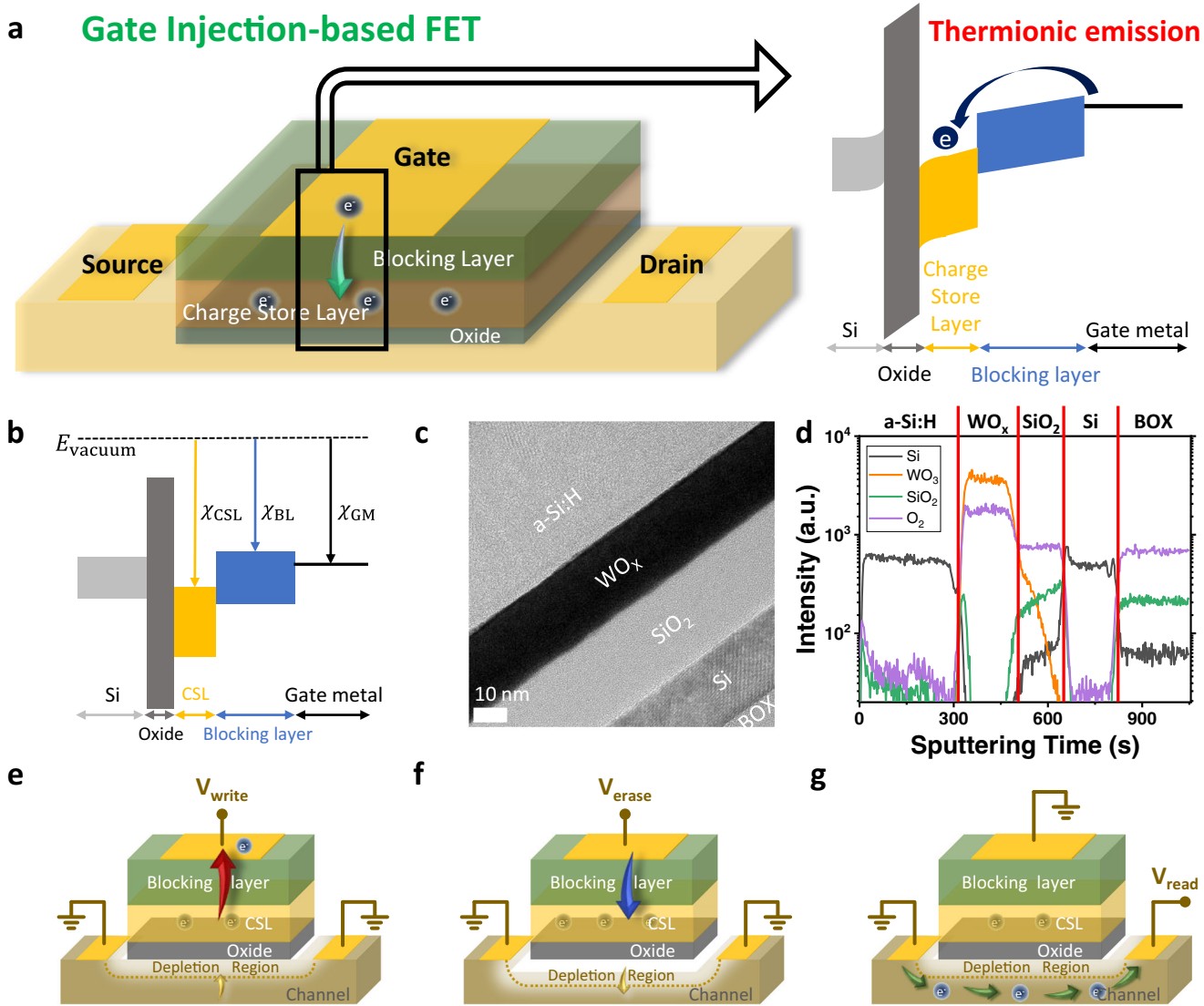

**Fig. 1 | Structure and operation principle of GIFET. a** Schematic and band diagram of GIFET. The charge is injected from or extracted to gate metal by thermionic emission. **b** Flat band diagram of GIFET. The electron affinity of the charge store layer (CSL) is higher than the blocking layer. **c** TEM image of the cross section of the gate stack. **d** SIMS data of the gate stack of GIFET. Schematic of GIFET during write (**e**)/erase (**f**)/read (**g**) operation, respectively. The depletion region of the channel is changed following charges in the CSL.

due to its lower dielectric constant than a-Si:H if they are in the same thickness. Therefore, the voltage drop occurs through the interfacial layer, and the barrier height between $WO_x$ and a-Si:H decreases. Accordingly, the conduction mechanism between CSL and blocking layer converted from thermionic emission to field emission or trap assisted tunnelling through $SiO_x$ layer (see Supplementary Fig. 4b). Supplementary Fig. 4c shows the Arrhenius plot of $\ln(I/T^2)$ versus $q/kT$ for the device with the interfacial layer. The zero slope of the graph shows that the disappearance of the Schottky barrier, and the linear relationship between $\ln(I/V^2)$ and $1/V$ with the interfacial layer at room temperature in Supplementary Fig. 4d implies the changed conduction mechanism is F-N tunnelling[50].

Consequently, the LTP-LTD characteristics of both the GIFET device with the $SiO_x$ layer between CSL and blocking layer (field emission dominant) and without the layer (thermionic emission dominant) were estimated to examine the relation between linearity and conduction mechanism for charge transfer through the blocking layer. As displayed in Supplementary Fig. 4e, the device with $SiO_x$ interfacial layer loses linear conductance update property compared with the device without interfacial layer under the same pulse train

(write: 2.5 V, 500 μs, erase: −3 V, 500 μs, read: 1 V, 500 μs), especially during LTP.

Figure 2c is the LTP-LTD characteristic of the GIFET observed with the 1000 potentiation (500 μs, 5 V)–1000 depression (500 μs, −3.3 V) gate pulse trains (see Supplementary Fig. 5 for pulse information). As shown in Fig. 2c, the device has high linearity with low asymmetric ratio[32,51] (see Supplementary Fig. 6 and Note 1). Conductance ratio $G_{max}/G_{min}$ around 10 is achieved, which was reported as the on/off ratio value for achieving high performance in ANN task[33]. Furthermore, 1000 analog conductance levels are more than enough for high accuracy. Figure 2d is enlarged conductance update from the part of the data in Fig. 2c. Each figure illustrates potentiation or depression of the weight with 100 switching pulses. As shown in Fig. 2d, the device shows stable linear conductance updates in the entire conductance state, which means it has similar conductance changes with the same number of pulses.

Program operation time of GIFET is practicable to be reduced while updating conductance linearly by controlling the pulse condition to optimize appropriate synaptic properties, such as on/off ratio and linearity for neuromorphic applications. In

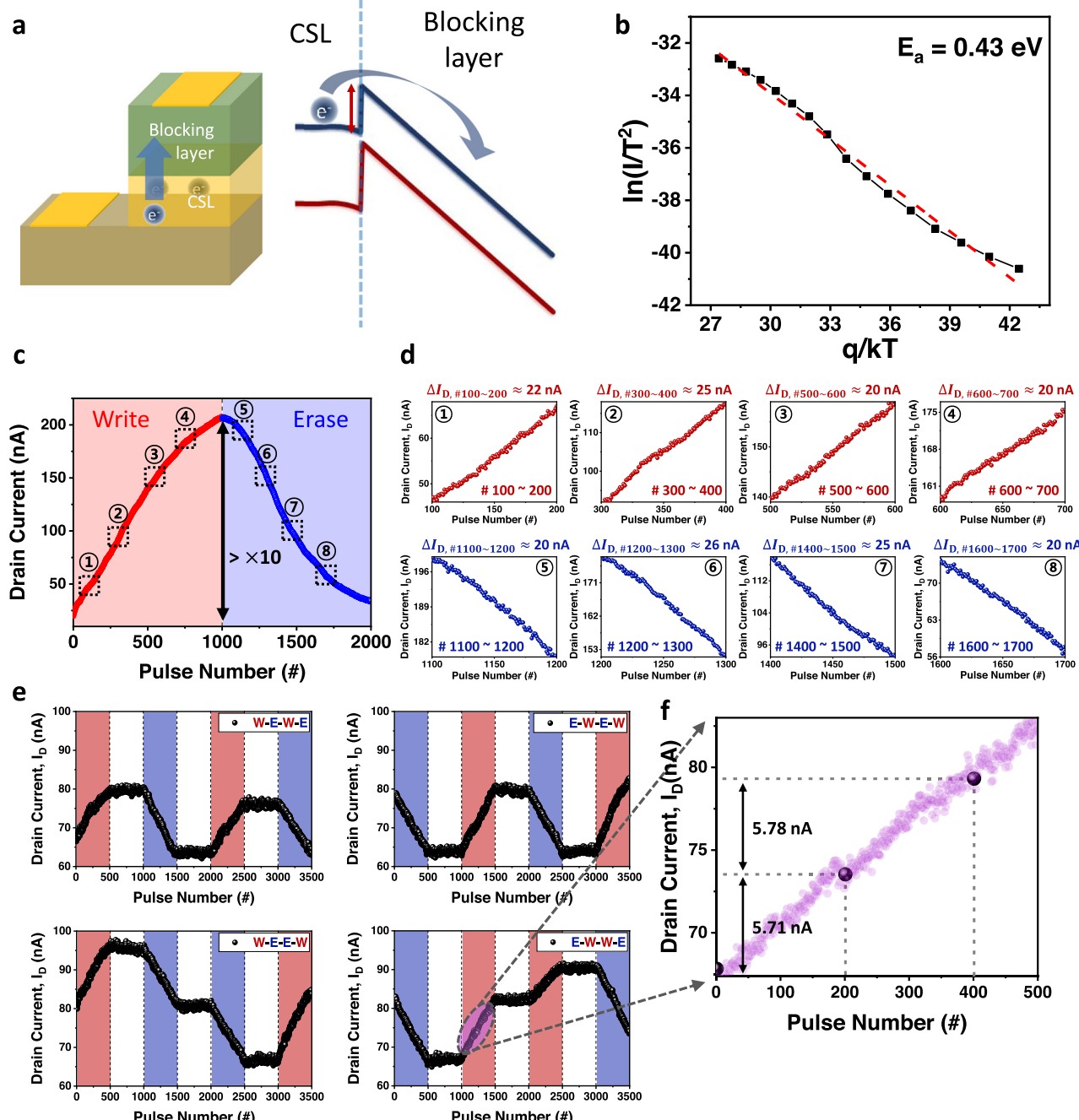

**Fig. 2 | The linearity of GIFET and its relationship with the conduction mechanism. a** Schematic of the GIFET gate stack without gate oxide and band diagram of CSL and blocking layer during write operation. The barrier exists between the CSL and the blocking layer, and the charge in the CSL is extracted over the barrier. **b** The Arrhenius plot measured from $WO_x$/a-Si:H stack without the $SiO_2$ layer between the channel and the stack from 273 K to 423 K. **c** The LTP-LTD characteristic of GIFET under 1000 write (5 V, 500 μs)−1000 erase (−3.3 V, 500 μs) consecutive pulse scheme (1 V, 50 μs read voltage and 50 μs between each pulse was used). **d** Linear write (red sphere)/erase (blue sphere) update of the LTD-LTD during every 100 pulses in specific ranges matched the numbers shown in **c**. Almost similar drain current increased/decreased within the same number of pulses. **e** The LTP-LTD characteristics of the GIFET under arbitrary pulse trains (haphazard write/ erase pulse) consist of 500 write (1.4 V, 500 μs)/erase (−2.5 V, 500 μs)/hold (0 V, 500 μs) gate pulses with a read pulse (1 V, 200 μs) on the drain 200 μs after each gate pulse (red: write operation, blue: erase operation, white: hold operation). **f** Stable linear update values of data in **e** (E-W-W-E) ($\triangle I_D$ = 5.71 nA, 5.78 nA).

practical machine learning applications, the program operation time of synaptic devices is considered the main parameter of the system speed. Therefore, minimizing the operation time is important, and the shortest program operation time of the GIFET for linear weight update was determined to be 200 μs (see Supplementary Fig. 7).

Figure 2e presents the linear conductance update with arbitrary pulse trains. Pulses consisting of 500 write pulses (1.4 V,

500 μs), 500 hold pulses (0 V, 500 μs) and 500 erase pulses (−2.5 V, 500 μs) were applied to the gate. For the read process, a read pulse (1 V, 200 μs) was applied to the drain terminal. As shown in Fig. 2e, during the hold process, current change has not been observed, which means that the data is well preserved. It is important for synaptic devices to maintain nonvolatile states at different intermediate conductance levels for neuromorphic computing applications. Moreover, the device has

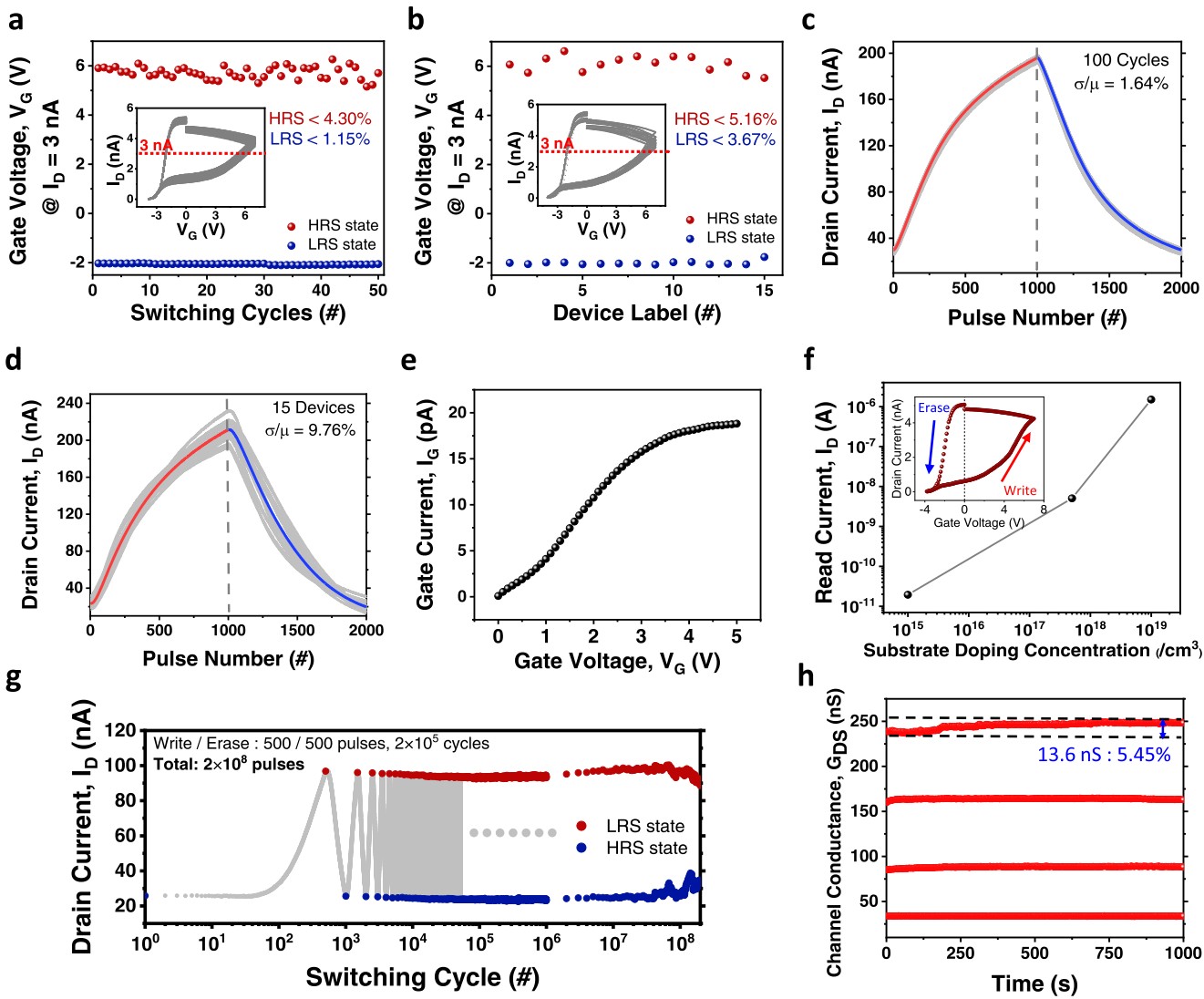

**Fig. 3 | Measured GIFET data for high performance in a crossbar-array structure.** Selected gate voltages based on reaching a constant current (3 nA) in $I_D$-$V_G$ characteristics. Gate voltage was double swept from −4 V to 7 V, and 50 cycles in a single device (**a**) and 15 different devices with the same gate width and length (**b**) were measured. Insets are based on data from Supplementary Fig. 12. **c** Repeated LTP-LTD characteristics on a single GIFET device with 1000 potentiation–1000 depression (5 V/−3.3 V, 500 μs). **d** LTP-LTD characteristics for 15 different devices with 1000 potentiation–1000 depression (5 V/−3.3 V, 500 μs). The cycle-to-cycle and device-to-device variations of GIFET are 1.64% and 9.76%, respectively. **e** $I_G$-$V_G$

characteristic of the GIFET. **f** LRS current level of GIFET in $I_D$-$V_G$ characteristic (inset) according to channel doping concentration. **g** The endurance of GIFET over $2 \times 10^5$ switching cycles ($2 \times 10^8$ pulses). Each switching cycle is composed of 500 write pulses with 5 V, 200 μs and 500 erase pulses with −5 V, 200 μs. **h** Retention characteristic of GIFET. Erase pulses (−3.3 V, 500 μs) were applied to the gate to reach a certain conductance level, and read pulses (1 V, 50 μs) were applied to the drain every 1 s after the erase pulse train. A 5.45% conductance value change was observed after 1000 s.

constant conductance change under repeated write/erase pulse train, as magnified in Fig. 2f. This data implicates that the weight stored in the device can be manipulated under an identical pulse scheme, which helps soften the burden of the peripheral circuit[12,37].

Next, the relationship between operation pulse amplitude/duration and conductance change per pulse was investigated as shown in Supplementary Fig. 8. It shows the relationship between the pulse amplitude and current change at different pulse durations. GIFET can control the linear conductance update with small current change per pulse in various pulse scheme (see Supplementary Fig. 9–11). In other words, the GIFET shows stable characteristics for controlling the pulse scheme for neuromorphic computing applications, and has the advantage of being customizable to fit the needs of other applications.

## Operational stability as a synaptic device

Besides the linearity in the LTP-LTD, to obtain high performance in the large-scale crossbar-array structure, it is highly required to satisfy various characteristics such as uniformity, $G_{max}/G_{min}$ ratio, low power consumption, endurance, and retention simultaneously[9,32,33,35,52]. To be integrated as a large-scale crossbar array, spatiotemporal uniformity is one of the essential properties[28,52].

First, to investigate the spatio-temporal uniformity of the GIFET, we assessed $I_D$–$V_G$ characteristics of the GIFET by gate voltage sweeping with constant read voltage on the drain. Figure 3a, b present gate voltage at a specific drain current ($I_D$ = 3 nA) from repeated cycles on a single device for cycle-to-cycle variation and from 15 different devices for device-to-device variation, respectively. In these figures, cycle-to-cycle variation was observed as 4.30% at HRS and 1.15% at LRS (σ/μ), while device-to-device variation was measured as 5.16% at HRS

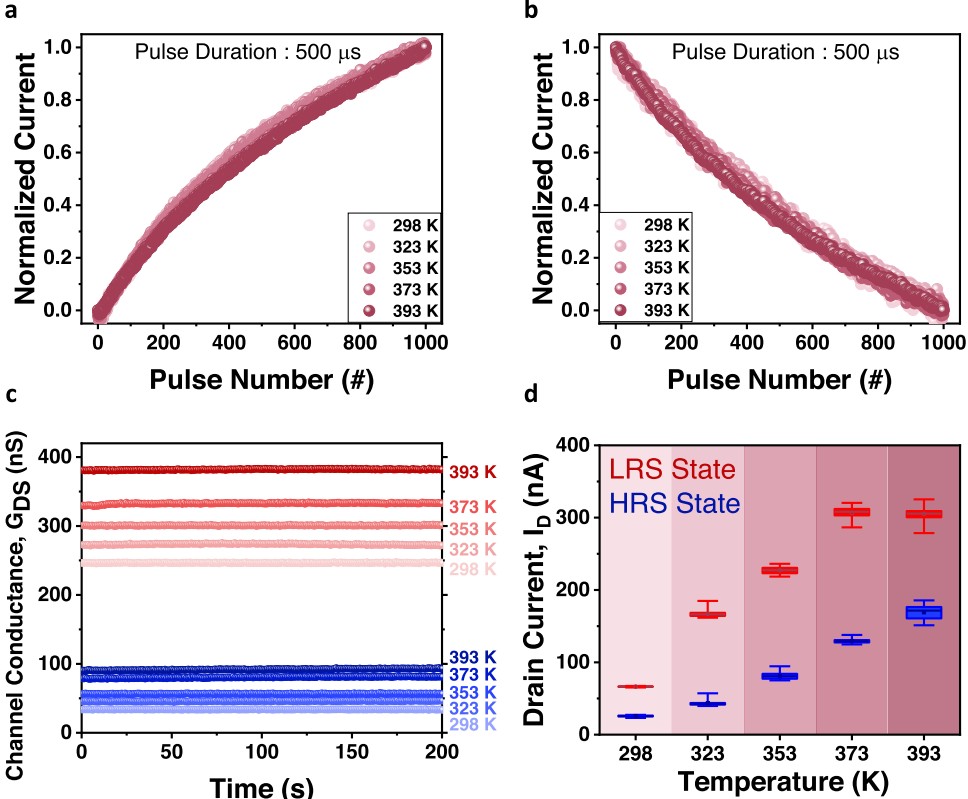

**Fig. 4 | Robustness of the GIFET to temperature variations.** Linearity variation in various temperatures from 298 K to 393 K with **a** 1000 potentiation pulses (4 V, 500 μs) and **b** 1000 depression pulses (−4 V, 500 μs). **c** Measurement of retention characteristics of two conductance states at various temperatures from 298 K to 393 K. Erase pulses (−3.3 V, 500 μs) of 100 (red sphere) and 1000 (blue sphere) were applied in the initial state, respectively, and read operations (1 V, 50 μs) were performed every 1 s after the erase pulse train. **d** Comparison of endurance characteristics at different temperatures after $10^5$ switching cycles ($2 \times 10^6$ pulses). The whisker represents minimum and maximum values and the box range indicates drain current values in the 25% to 75% percentile.

and 3.67% at LRS (σ/μ) (ref. Supplementary Fig. 12). In addition, the repeated LTP-LTD characteristics on a single device and several different devices were observed with 1000 potentiation (5 V, 500 μs,)–1000 depression (−3.3 V, 500 μs) gate pulse trains, as shown in Fig. 3c, d. The LTP-LTD characteristics of the device in Fig. 3c presented a low variation of 1.64% (σ/μ) for 100 repeated cycles. The device-to-device variation was experimentally measured on 15 devices and it showed 9.76% (see Fig. 3d). The standard deviation of nonlinearity based on the LTP-LTD characteristics of 15 devices is also calculated in Supplementary Fig. 13. The nonlinearity during potentiation/depression in Supplementary Fig. 13a, b was fitted using the method from Supplementary Note 2. Each spatial variation in the results of the above $I_D$-$V_G$ and LTP-LTD characteristics was slightly higher than each temporal variation because of the Si channel thickness variation during fabrication (see Methods section). This structure shows uniform switching because it utilizes a large population of electrons, minimizing the effect of fluctuation or stochastic behavior of individual charged particles, instead of individual ion movement.

Figure 3e shows the $I_G$-$V_G$ characteristics of the GIFET. As observed, the gate current of the device was lower than 20 pA at 5 V gate bias, which means that power consumption for a write pulse with 500 μs pulse width is lower than 50 fJ. In addition, the power consumption during the read process in Fig. 3c is 5.54 pJ at the maximum conductance level. Notably, the read current level of the device can be modulated by controlling Si channel doping concentration (see Fig. 3f and Supplementary Fig. 14), indicating that we can tune the device operation speed and power consumption for specific applications such as edge computing processor and high-performance processor.

The endurance and retention of the device are also crucial for long-term and reliable neuromorphic computing applications[52]. To investigate the endurance of GIFET, we applied 500 consecutive potentiation pulses with an amplitude of 5 V and a width of 200 μs, followed by 500 consecutive depression pulses with an amplitude of −5 V and a width of 200 μs per switching cycle. We then read the change in state by drain voltage (1 V, 50 μs) at each switching cycle. As presented in Fig. 3g, the device achieves robust endurance ($\geq 2 \times 10^8$ pulses).

Figure 3h shows the data-holding ability of the GIFET. We observed a data loss of 5.45% (13.6 nS) of the updated conductance after 1000 s. Also, several intermediate conductance levels were maintained without severe degradation. It is essential for synaptic devices to maintain non-volatile states at various intermediate conductance levels in order to be used for neuromorphic computing applications. There is a tradeoff between retention, endurance, and linearity for weight update due to lowered barrier height of the blocking layer. The newly developed device improves endurance and linear conductance update while it loses data holding ability compared to conventional flash memory devices. Because the barrier height between the blocking layer and CSL can be controlled by the material stoichiometry and film quality[53,54], the device characteristic can be optimized for specific purposes through further engineering of processes and materials utilized for CSL and blocking layer.

## The Robustness of the GIFET to temperature variations
Figure 4a, b present the linear characteristics of the GIFET under varying temperatures. We confirmed the robustness of the linearity over a temperature change from 298 K to 393 K during 1000 potentiation (4 V, 500 μs) and 1000 depression (−4 V, 500 μs) operations. As shown in Supplementary Fig. 15, the linearity of conductance update is stable at all temperatures without severe degradation.

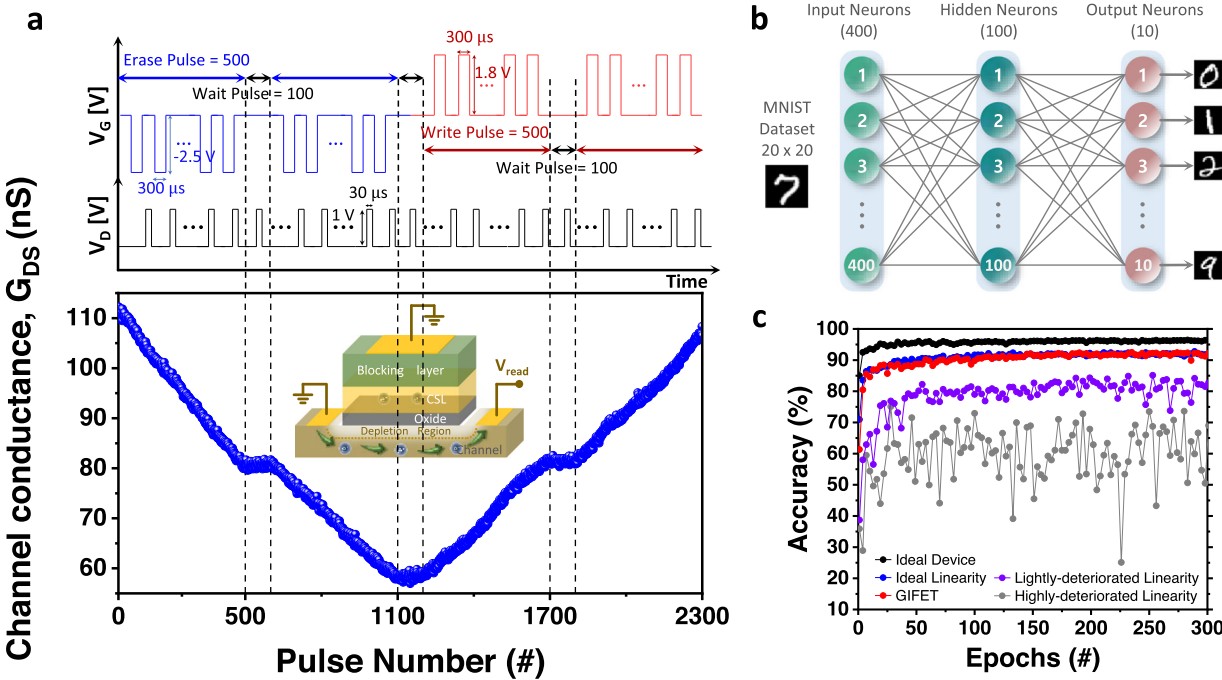

**Fig. 5 | GIFET simulation with MNIST dataset. a** The fast read speed of the GIFET. The graph shows the LTP-LTD characteristics of the GIFET cell with 1 V, 30 μs read pulses (write/erase with 1.8 V/−2.5 V, 300 μs). **b** The configuration of the multi-layer perceptron (MLP) artificial neural network for image classification simulation based on MNIST handwritten data. The parameters of the GIFET was reflected for each synapse. **c** MNIST image classification simulation to compare the GIFET characteristics with ideal device (software baseline simulation with ideal circumstances such as large $G_{max}/G_{min}$ ratio, ideal linearity, and no device variations), ideal linearity (reflecting all the characteristics of GIFET except linearity and device variations) and the devices with deteriorated linearity (nonlinearity of 3, 6 for lightly-, highly-deteriorated linearity, respectively).

For verifying the reliability of multi-level conductance states in high temperatures, the retention characteristics were measured for two states according to temperature from 298 K to 393 K as shown in Fig. 4c. Erase pulses (−3.3 V, 500 μs) of 100 (read sphere) and 1000 (blue sphere) were applied in the initial state, respectively, and read operations (1 V, 50 μs) were performed every 1 s. It was confirmed that only the conductance level (drain current level) increased as the temperature increased and that the conductance states remained unchanged for 200 s. This indicates long-term plasticity properties, and the proposed GIFET can hold data for online training, even at high temperatures.

Figure 4d and Supplementary Fig. 16 shows the endurance characteristics of GIFET. To investigate the robustness of the hardware, the endurance was measured under the same pulse conditions over a temperature change from 298 K to 393 K. The pulse train consists of ten consecutive potentiation pulses with an amplitude of 6 V and width of 500 μs, followed by ten consecutive depression pulses with an amplitude of −6 V and width of 500 μs. As presented in Fig. 4d and Supplementary Fig. 16, the device operates stably by holding its high-level and low-level states, without severe degradation after $10^5$ switching cycles (2×$10^6$ pulses in total). This indicates reliable endurance characteristics for highly frequent updates during online learning.

**ANN simulation with the performance of GIFET**

Figure 5a presents the read operation of the GIFET with 1 V, 30 μs pulse to read changed conductance with 1.8 V/−2.5 V, 300 μs update pulses. As presented in the graph, the read operation of GIFET can be conducted without applying gate bias by read pulses of 30 μs, which is comparable with that of conventional NAND flash[55]. Therefore, it has advantages in terms of low power consumption for dense array application, while NAND flash needs repeated processes that are determining on/off of the cell using threshold voltage with applying specific bias for reading current state.

To examine the performance of the GIFET for neuromorphic computing as a large-scale crossbar array structure, we simulated the device with the multi-layer artificial neural network using long-term plasticity characteristics directly extracted from measured data[33] (see Supplementary Fig. 17 and Note 4). The multi-layer perceptron ANN consists of an input layer with 400 nodes, a hidden layer with 100 nodes, and an output layer with 10 nodes, as shown in Fig. 5b. Each 400 input node represents each pixel of 20×20 MNIST handwritten data and this input data resulted in 10 output through 2-layer of vector-matrix multiplication and activation function. Ten output nodes mean the result of classification among 0-9 digits. Each weight of the synapse device was updated based on the stochastic gradient descent method with parameters of the GIFET such as nonlinearity, $G_{max}/G_{min}$ ratio, cycle-to-cycle variation, device-to-device variation, and applied pulse scheme to account for device non-ideality. 8000 random images per each epoch out of 60,000 training image set were utilized in the training process. After training, the system accuracy was evaluated with 10,000 MNIST images of a testing set. To inspect the influence of linearity on ANN performance, we conducted the simulations with varying linearity. Figure 5c presents the resulting accuracy graphs of MNIST classification by each epoch with the GIFET, the software baseline with ideal device, the device with ideal linearity and symmetry, and the device with deteriorated linearity (see Supplementary Fig. 18). As observed, the ideal device shows an accuracy of 96.78%, and the GIFET-based artificial neural network obtained an accuracy of

 

**Table 1 | Comparison with the various synaptic transistors for neuromorphic computing**

| | 57 | 44 | 35 | 32 | 58 | 41 | 59 | This work |
|---|---|---|---|---|---|---|---|---|
| The Number of Conductance States (LTP/LTD) | 100/100 | 60/60 | 64/64 | $2^{13}/2^{13}$ | 100/50 | 50/50 | 100/100 | 1000/1000 |
| Area [μm²] | 0.52 | 3 | 15,000 | 350 | N/A | N/A | 2000 | 100 |
| Power Consumption [fJ] | <400 | 30 | ≈60,000 | <20 | N/A | ≈160 | N/A | <50 |
| On/off Ratio | ≈3 | ≈2 | >10 | ≈$10^3$ | >10 | ≈2 | ≈9 | >10 |
| Program Voltage [V] | 2 | 1.2 | 2.7~4.3 | 4 | 3.5 | 2.5 | 3 | 1.8 |
| Program Time [ms] | 100 | 100 | 10 | 10 | 10 | 10 | 100 | 0.3 |
| Temporal Variation | N/A | N/A | 2.36% | N/A | N/A | <6.5% | N/A | 1.64% |
| Spatial Variation | N/A | N/A | 3.93% | N/A | N/A | <12% | N/A | 9.76% |
| Linearity (ideal = 1) (LTP/LTD) | 1.5/5.9 | 1.9/0.5 | *ISPP* | 1.51/−0.38 | 1.3/−0.3 | N/A | 0.96/−0.11 | 1.53/0.47 |
| Retention | N/A | N/A | $10^4$ s | 5000 s | N/A | 100 s | N/A | >1000 s |
| Endurance (HRS/LRS switching cycles) | N/A | N/A | >$10^5$ | >400 | N/A | >40 | N/A | >2×$10^5$ |
| Simulation Accuracy | 84.6% | N/A | 91.1% | 91.7% | 86.82% | 87.3% | 93.26% | 93.17% |
| CMOS Compatibility | No | No | Yes | Yes | Yes | No | No | Yes |

ISPP (Incremental Step Pulse Programming) is not identical pulse.

approximately 93.17% during 300 epochs, which is almost equivalent to that of an ideal linearity device, 93.45%. Furthermore, the device with lightly- and highly-deteriorated linearity exhibited degraded training results as maximum accuracy of 85.13% and 70.56%, respectively. Compared to other synaptic parameters of the same algorithm, the GIFET shows high accuracy with fair comparison including number of conductance states, nonlinearity, on/off ratio and spatio-temporal variation. (see Supplementary Table 1). These results demonstrate the importance of the linear conductance update for neuromorphic computing on a large-scale crossbar array and suggest that the GIFET has enough linearity.

## Discussion

In summary, we developed a three-terminal synapse device for enhancing linearity on LTP-LTD based on field-effect to control channel conductance by stored charge in CSL, which is injected from or extracted to gate metal based on thermionic emission. The effect of the conduction mechanism between CSL and gate metal through the a-Si:H blocking layer on linear conductance update was investigated by comparing the device with and without interfacial layer and observing the gate current through the blocking layer of each device under varying temperatures. The thermionic emission-based GIFET reported linear conductance update while the device with interfacial layer presented nonlinearity. Furthermore, GIFET shows superior properties such as number of conductance states, area, power consumption, on/off ratio, operating voltage, programming time, spatio-temporal variation, linearity, retention, endurance, simulation accuracy and CMOS compatibility (see Table 1), since the mechanism based on electron movement employs the flash memory structure. In addition, low spatio-temporal variation, reliable endurance and retention, and low power consumption of the GIFET support that the device is qualified for the large-scale crossbar array to conduct neuromorphic computing. Moreover, all the processes and materials utilized for the GIFET fabrication were CMOS compatible, which suggested low-cost and fast integration with the conventional system. Artificial neural network simulation based on MNIST dataset with the parameters extracted from GIFET measurement data shows high accuracy of 93.17%, which implies the possibility of AI acceleration with GIFET-based large-scale crossbar array.

## Methods

### Device Fabrication

The Si top layer of SOI wafer with 145 nm thickness oxidized by thermal furnace and the oxide was removed by hydrofluoric acid to reduce the thickness of the Si layer, remaining 20 nm. Ion implantation (Dose $2 \times 10^{13}$ cm⁻² 7.5 KeV, Phosphorus) and annealing (1273 K, $N_2$ atmosphere, 1 min) was conducted. Each cell on buried oxide were designed with 5 μm channel width by mesa pattern lithography and Reactive Ion Etch (RIE) with $SF_6$ and Ar gas. Silicon dioxide 25 nm was deposited as gate oxide using Plasma Enhanced Chemical Vapor Deposition (PECVD). $WO_x$ was deposited with RF sputtering using $WO_3$ sputtering target (Kurt J. Lesker, USA). The sample was annealed in Rapid Thermal Annealing system with $O_2$ atmosphere, 573 K to enhance stoichiometry of $WO_x$ layer (see Supplementary Fig. 19 and Note 5). The $WO_x$ layer was designed with 20 μm gate length by lithography and RIE etch with $SF_6$ and Ar gas. Hydrogenated amorphous silicon 50 nm was deposited using PECVD and designed by lithography and RIE etch. Silicon dioxide on source and drain was removed by wet etch with BOE. Lastly Ti/Au (10 nm/50 nm) was deposited as metal pad for source, drain, and gate.

### Electrical Measurement

Parameter analyzer (Keithley 4200A-SCS) with the conventional probe station was used to measure $I_D$-$V_G$ characteristic of the GIFET by gate voltage sweeping during applying read voltage on drain. The resolution of sweep gate bias was 0.05 V. The analog conductance update under successive identical pulse train was measured using parameter analyzer and Pulse Measurement Unit (PMU), which allow setting pulse width and amplitude on gate and drain intentionally. Read pulses applied to drain after every write or erase process to read conductance state.

Data acquisition system (USB-6363, National Instrument) and current preamplifier (DL instruments, Model 1211) were utilized to measure the endurance of the GIFET. The pulse magnitude and width were modulated through MATLAB® code. The repeated switching cycle applied to the GIFET through DAQ and output current flowed to preamplifier under drain read voltage. Current was measured by averaging the output current over specified duration. Each switching cycle was similar to what we utilized in measurement of analog conductance update with parameter analyzer and PMU. LRS and HRS of each switching cycle were extracted.

### Conduction mechanism analysis

To investigate the conduction mechanism through gate stack, I-V characteristic was measured at varied temperature under vacuum condition (-10⁻² Torr). Keithley 236 Source Measurement Unit (SMU) driven by LabVIEW was utilized to apply voltage and measure current with cryogenic probe station (ModuSystems, Inc).

## MNIST simulation based on GIFET array

The simulation of the GIFET based crossbar array was conducted based on "NeuroSim+". The neural network was composed of three layers to conduct supervised learning with back propagation. The input layer had 400 nodes for 20 × 20 pixels of binary MNIST image and the hidden neuron had 100 nodes, while the output neuron had 10 nodes for results of classification, representing 0-9 digits. Stochastic gradient decent was used for weight update. The gradient of the cost function for the neural network parameters was computed using a stochastic gradient descent algorithm. Stochastic gradient decent randomly samples examples from the training dataset for each epoch to compute the gradients. Therefore, it is usually much faster and widely used for the training process[56].

The simulation consists of two parts: the synaptic array and peripheral circuitry. The peripheral circuit includes a switch matrix, crossbar WL decoder, MUX decoder, analog-to-digital read circuit, adder, and shift register. The device parameters of GIFET such as set voltage, pulse width, min/max conductance, nonlinearity, cycle-to-cycle variation, and device-to-device variation are utilized to perform the simulation.

## Data availability

The data that support the findings of this study are available from the corresponding author upon reasonable request.

## Code availability

The codes used for the simulations are available from the corresponding author upon reasonable request.

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

## Acknowledgements

This work has partially supported by the R&D program of Korea Evaluation Institute of Industrial Technology (KEIT) grant funded by the Korea government (Ministry of Trade, Industry, and Energy) (20003789) and by the R&D programs of National Research Foundation of Korea (NRF) grant funded by the Korea government (Ministry of Science and ICT) (2018R1A2A3075302, 2019M3F3A1A02072336, 2020M3F3A 2A01085755, 2020M3F3A2A01082592, 2021M3F3A2A01037858, 2022M3F3A2A01072851, and 2022M3I7A2078273), and by Nanomedical Devices Development Project of National Nano Fab Center (CMS2103M001).

## Author contributions

S.S., B.K., D.K., and S.P. contributed equally to this work, and S.C. directed the team. T.R.K., S.O.P., and S.C. designed the basic concept of the idea. S.S., B.K., D.K., and S.P. planned and performed the experiments. S.S., B.K., S.P., D.K., T.P., H.S., M.S.K., and Y.K.C manufactured the device. S.S., B.K., D.K., S.P., H.J., M.S.K, and Y.K.C measured the device and analyzed the data. J.P. conducted array simulation. S.S., B.K., D.K., and S.C. wrote the manuscript.

## Competing interests

The authors declare no competing interests.

## Additional information

**Correspondence and requests** for materials should be addressed to Shinhyun Choi.

