## [Peer Review File · Nature Communications]

Reviewers' Comments:

Reviewer #1:

Remarks to the Author:

In this manuscript, the authors propose a CMOS compatible field-effect synaptic transistor which has the advantages of linear conductance update, high uniformity and low power consumption. The authors also investigate the underlying mechanism leading to the linear conductance modification and demonstrate the superior synaptic properties based on thermionic emission for high-performance machine learning tasks. Here are some of the suggestions to help improve the article:

- (1) Compared to the reliability concerns of memory application, the concerns of neuromorphic computing application focus on the accurate conductance states. Hence, it's important for synaptic devices to maintain non-volatile states at different intermediate conductance levels.
- (2) In designing the GIFET, the authors choose WO_x and a-Si:H as the CSL and blocking layer respectively. What is the effect of the thickness of these two layers on device performance? How to choose the appropriate material thickness?
- (3) In the manuscript, the authors use 300 and 500 μ s as the write/read operation voltage. In practical machine learning applications, the write/read operation time of the synaptic devices have a very important effect on the system speed. Did the authors explore the shortest write/read operation time of the GIFET for linear weight update?
- (4) In the experimental section, the authors mention that the simulation of the GIFET based crossbar array was conducted based on "NeuroSim+". More details about how to model crossbar array and perform network simulation should be provided in the article.
- (5) As a transistor based device, the authors should characterize the output curve to reveal more electrical properties and details about the device.
- (6) Description error. In Supplementary Note 4, it is the XPS data in Figure S8 the authors are talking about instead of the content in Figure S6. Please check the manuscript carefully.

Reviewer #2:

Remarks to the Author:

Comments to the author:

Major comments :

1. Memories work on the non-linearity of program vs retention e.g. ms program vs 10 years (1e8s) retention is an electron injection rate modulation of 1e11. This is provided by FN tunnelling. Choosing thermionic emission is regressive as it does not have sufficient non-linearity. The improvement in a specific performance metric i.e. linearity is not unexpected but the degradation retention does not move the "main" trade-off. This is the central challenge. Achieving one specification while neglecting another key specification is not interesting. Should this be compared against volatile memory like DRAM? In well-established DRAM, linearity is excellent - much better than the present device. It works at 3.3V or even 1.8V as opposed to the present device. So without retention, the idea does not remain compelling.
2. The author claims that the thermionic emission-based GIFET improves linearity, which leads to an accuracy of 93.03% in MNIST data classification. Similar accuracy results are also demonstrated using other synaptic devices available in the literature [1-3] but at the expense of more complex peripheral circuitry. This claim needs to be supported using a quantitative comparison between the SOTA architectures. If the impact is in the accuracy, then accuracy improvement w.r.t previous works should be shown; if the impact is in peripheral circuitry, then improvement w.r.t previous works on those lines should be shown.
3. The temperature variability of GIFET is unexplored in the manuscript. Please add necessary figures indicating temperature dependence on states, retention time, and endurance time. The effect of temperature variations on linearity, retention time, and writing speed will require a compensatory circuit leading to the increased complexity of the peripheral circuitry. Robustness of hardware to temperature variations during operation needs to be shown since the mechanism is now temperature-sensitive as opposed to the typical FN tunnelling.
4. The authors show a dynamic range of 10x with 1000 pulses used to achieve that. FN tunnelling-based transistor memories with a much higher dynamic range can always reduce their dynamic

range to improve linearity in the reduced dynamic range. These devices have much better retention [4]. How do authors motivate this change of writing mechanism for improved linearity in this case?

5. The main manuscript should include a comparison with other synaptic devices, clearly highlighting improved factors/advantages. (Critical parameters to be included in the comparison are Area, Power, Linearity, Programming time, Retention time, Accuracy benefits, temperature dependence, and variability.)

6. At its present state, the synapse area is $5\mu\text{m} \times 20\mu\text{m}$, which is very large compared to SOTA synaptic devices. The gate stack is 75 nm thick. The device should not be scalable below 75 nm in the horizontal direction without incurring severe fringing fields, interFG interference, and short channel effects. The smaller size is an essential parameter for achieving a higher packing density of synapses in neural circuits. Is there any possibility of scaling the device, and will that affect the conduction mechanism and linearity? Without this analysis, it is difficult to assess the relevance of this device.

Minor comments:

1. The author has not shown the effect of device-to-device variability on the Linearity of the device. Please provide a figure similar to 3c and 3d indicating device-to-device variability. Also, consider device-to-device variability in the NeuroSim+ simulation to calculate the accuracy of MNIST classification.

2. The authors claim that the use of two-terminal devices as synapses suffers from device-to-device variation and operation stochasticity. However, the proposed GIFET also suffers from device-to-device variability, as shown in fig. 3b. A quantitative comparison is necessary to indicate reduced variability.

3. The author is requested to add plots indicating the relationship between write pulse amplitude and conductance change per pulse. Also, add a similar graph showing a relationship between the write pulse duration and conductance change per pulse.

4. How does lower retention time affect the training? What is the plan for inference which requires long-term non-volatile weight storage? (As the conduction mechanism is changed to thermionic for which the barrier height between CSL and Blocking layer is decreased)

5. Extend plot of I_d vs. pulse number till I_d saturates during the writing operation and erasing operation. It shall provide a maximum/minimum capacity of the charge storage layer and maximum dynamic range.

6. Check figure S1b, the band diagram.

7. Page 3, line 64: correct the spelling to Fowler-Nordheim tunnelling.

[1] S. Liu et al., "A TaOx-Based Electronic Synapse With High Precision for Neuromorphic Computing," in IEEE Access, vol. 7, pp. 184700-184706, 2019, doi: 10.1109/ACCESS.2019.2961166.

[2] X. Sun, P. Wang, K. Ni, S. Datta and S. Yu, "Exploiting Hybrid Precision for Training and Inference: A 2T-1FeFET Based Analog Synaptic Weight Cell," 2018 IEEE International Electron Devices Meeting (IEDM), 2018, pp. 3.1.1-3.1.4, doi: 10.1109/IEDM.2018.8614611.

[3] S. Lee et al., "Operation Scheme of Multi-Layer Neural Networks Using NAND Flash Memory as High-Density Synaptic Devices," in IEEE Journal of the Electron Devices Society, vol. 7, pp. 1085-1093, 2019, doi: 10.1109/JEDS.2019.2947316.

[4] V. Bhatt, S. Shrivastava, T. Chavan and U. Ganguly, "Software-Level Accuracy Using Stochastic Computing With Charge-Trap-Flash Based Weight Matrix," 2020 International Joint Conference on Neural Networks (IJCNN), 2020, pp. 1-8, doi: 10.1109/IJCNN48605.2020.9206631.

Reviewer #3:

Remarks to the Author:

The manuscript reports a gate injection based field-effect transistor with superb synapse performance in artificial neural networks. The transistor shows high linearity on long-term plasticity with the mechanism of thermionic emission. The work is valuable to realizing CMOS-compatible neuromorphic computing. Before publication there are several problems that need to be addressed by the authors.

1. What are the exact values of the energy levels in Fig. 1b? The authors need to provide experimental evidence or references for the band diagram.
2. Does the transistor work in a depletion mode? If so, how is the depletion region in the channel formed?
3. Why the absolute value of writing voltage is not equal to that of the erasing voltage? It is interesting that the absolute value of writing voltage is higher in Fig. 2c and lower in Fig. 3b. Could the authors clarify the reason for choosing the writing and erasing voltage?
4. Could the authors explain the stochastic gradient descent method more clearly?
5. The authors explain the conduction mechanism by I-V characteristic measurements in the temperature range of 339 K ~ 413 K. What temperature do the authors measure the LTP-LTD characteristics at? Whether is the conduction also dominated by thermionic emission at this temperature?
6. In Figure 2e the drain currents are almost the same after 500 write/erase pulses except for the W-E-W-E process. Please clearly discuss the underlying mechanisms?
7. For Figure 2c, e and 3f the authors have measured the LTP-LTD characteristics and endurance by using different voltages of write/erase pulses. Why do the authors not keep the voltages consistent?
8. The devices with lightly- and highly-deteriorated linearity would exhibit degraded training results of artificial neural networks. What about the linearity of the device with a different doping concentration of the Si channel?

Response Letter to Reviewers' Comments

We sincerely appreciate the reviewers for investing their valuable time and effort on reviewing our manuscript and providing insightful comments and suggestions to help further improve the quality of our work. Considering the reviewers' evaluations, we have made a point-by-point response to the reviewers' comments, and revised our manuscript to improve the clarity of our work. We believe we have addressed the reviewers' comments and now the paper is more rigorous in content and clearer in presentation. We have also revised grammar and expression improving clarity. Based on the responses below, we have appended seven figures and one table in the revised manuscript, thirty-three figures and two tables in the Supplementary Information, to address the comments. Our point-by-point responses to the reviewers' comments are as follows.

Reviewer #1 (Remarks to the Author):

In this manuscript, the authors propose a CMOS compatible field-effect synaptic transistor which has the advantages of linear conductance update, high uniformity and low power consumption. The authors also investigate the underlying mechanism leading to the linear conductance modification and demonstrate the superior synaptic properties based on thermionic emission for high-performance machine learning tasks. Here are some of the suggestions to help improve the article:

Response: We sincerely appreciate reviewer #1 for the constructive comments on our work and also for pointing out the important insight to emphasize the results. Our detailed responses to the reviewer's technical comments are provided below.

Comment #1:

Compared to the reliability concerns of memory application, the concerns of neuromorphic computing application focus on the accurate conductance states. Hence, it's important for synaptic devices to maintain non-volatile states at different intermediate conductance levels.

Response: We appreciate the reviewer for the helpful comment. We agree that maintaining non-volatile states at various intermediate conductance levels is essential for synaptic devices. The multi-conductance levels were tested, as shown in Fig. R1. The erase pulse train (-10 V, 1 ms) was applied 500 to 5,000 times to the gate to set 10 different multilevel states. Several intermediate conductance levels were observed without severe degradation over 100 s from our proposed device (read condition: 1 V at drain with 1 s interval).

The conductance level has changed slightly (7.15% changes in 1,200 s) owing to the charge leakage, which could be caused by the contamination during device fabrication. By optimizing the device fabrication process and investigating suitable materials in the gate stack, we believe that the retention at each multilevel state can be improved for neuromorphic computing applications.

In order to reflect the reviewer's suggestion and emphasize the properties of synaptic devices, we revised the sentences in the manuscript on page 14, line 265:

“Figure 3h shows the data holding ability of the GIFET. We observed a data loss of 7.15% (5 nS) of the updated conductance after 1,200 s. In addition, several intermediate conductance levels of the GIFET were observed without severe degradation as shown in Supplementary Fig. 15. It is also essential for synaptic devices to maintain non-volatile states at various intermediate conductance levels in order to be used for neuromorphic computing applications.”

We also appended Fig. R1 in the Supplementary Information as Supplementary Fig. 15.

Fig. R1 Retention characteristic at intermediate conductance states. The erase pulse train (-10 V, 1 ms) was applied 500 to 5,000 times to the gate to set 10 different multilevel states. The data were collected at 1 V with 1s interval for 100 s.

Comment #2:

In designing the GIFET, the authors choose WO_x and a-Si:H as the CSL and blocking layer respectively. What is the effect of the thickness of these two layers on device performance? How to choose the appropriate material thickness?

Response: We appreciate the reviewer for the constructive comment on our work. We

investigated the effect of the thickness of the devices by changing the thickness of each gate. In this study, we used a 25nm WO_x charge store layer (CSL), and a 50 nm a-Si:H blocking layer as the gate stack. The thickness of the CSL and blocking layer affects the energy band bending through the layers, resulting in the change of the operation voltage and current ranges. Therefore, it is necessary to select an appropriate stack thickness according to the doping profile and the thickness of the silicon channel.

The electrons stored in the WO_x layer induce a depletion region in the channel. I-V characteristic measurements of the GIFET with various thicknesses of WO_x are shown in Fig. R2a. A lower current level was observed for thinner WO_x layers. The thin WO_x layer results in a higher electron concentration at the interface between the WO_x and the bottom SiO₂. This strongly depletes the electrons in the channel, resulting in a low current.

The blocking layer (a-Si:H) determines the operating voltage of a GIFET. Figure R2b shows the effect of blocking layer thickness on the threshold voltage shift. If the thickness of the blocking layer increases, the total capacitance between the channel and the gate metal decreases. To form a depletion region in the silicon channel, a higher negative voltage is required for a thicker blocking layer. As a result, the threshold voltage shifted to the negative direction as the thickness increased.

Fig. R2 Electrical characteristics of GIFET in different stack thickness. a I_D - V_G characteristics of GIFET in three different WO_x layer thicknesses. **b** Threshold voltage shift as a function of thickness of a-Si:H layer. The thickness of the silicon channel and WO_x layer are 20 nm and 25 nm, respectively.

Comment #3:

In the manuscript, the authors use 300 and 500 μ s as the write/read operation voltage. In practical machine learning applications, the write/read operation time of the synaptic devices have a very important effect on the system speed. Did the authors explore the shortest write/read operation time of the GIFET for linear weight update?

Response: We appreciate this valuable comment. We agree that minimizing the program/read operation time is necessary because it directly affects the overall system speed of various machine learning applications.

As shown in Fig. R3, the shortest program/read operation time of the GIFET for the linear weight update was analyzed to be 200 μs /20 μs . To investigate the shortest read operation time, the program operation time was fixed at 200 μs . If the read operation time is shorter than 20 μs , the drain current is not read correctly due to the current overshooting problem during measurement as shown in Fig. R3a. Unfortunately, the available minimum pulse width for the current measurement system (Keithley 4200A-SCS and 4225-PMU with 4225-RPM) in the measure range of 1 μA is 20 μs . As the reviewer suggested, the smallest read operation time will be investigated as a future work.

The shortest write time was examined as shown in Fig. R3b. If the program operation time is shorter than 200 μs , regardless of the number of pulses, GIFET does not reach the minimum current level and sacrifices its on/off ratio as shown in Fig. R3b.

Our device has a faster program/read operation time than those of various synaptic transistors (see Table R1). GIFET has a structure similar to that of the NAND flash memory, but has the advantage of a faster write speed than conventional NAND flash^{R1}.

To reflect the reviewer’s comment, we added the following sentences in the revised manuscript on page 10, line 191:

“Program operation time of GIFET is practicable to be reduced while updating conductance linearly by controlling the pulse condition to optimize appropriate synaptic properties, such as on/off ratio and linearity for neuromorphic applications. In practical machine learning applications, the program operation time of synaptic devices is considered the main parameter of the system speed. Therefore, minimizing the operation time is important, and the shortest program operation time of the GIFET for linear weight update was determined to be 200 μs (see Supplementary Fig. 7).”

We also appended Fig. R3b into the Supplementary Information as Supplementary Fig. 7.

Fig. R3 Exploration of the shortest operation time of the GIFET. a Program operation characteristic under various read pulse duration. The shortest read operation time using the

current measurement system is 20 μ s. If the read operation time is shorter than 20 μ s, the drain current cannot be read correctly due to the overshooting problem. **b** Program operation characteristic under various program pulse duration. The shortest program operation time is investigated as 200 μ s under fixed read time. If the program operation time is shorter than 200 μ s, regardless of the number of pulses, GIFET is hard to be programmed into minimum current level as shown in the inset.

	[R18]	[R19]	[R20]	[R21]	[R22]	[R23]	[R24]	This work
The Number of Conductance States (LTP / LTD)	100 / 100	60 / 60	64 / 64	2 ¹³ / 2 ¹³	100 / 50	50 / 50	100 / 100	1,000 / 1,000
Area [μ m ²]	0.52	3	15,000	350	N/A	N/A	2,000	100
Power Consumption [fJ]	< 400	30	\approx 60,000	< 20	N/A	\approx 160	N/A	< 50
On/off ratio	\approx 3	\approx 2	> 10	\approx 10 ³	> 10	\approx 2	\approx 9	> 10
Program Voltage [V]	2	1.2	2.7 ~ 4.3	4	3.5	2.5	3	1.8
Program Time [ms]	100	100	10	10	10	10	100	0.3
Temporal Variation	N/A	N/A	2.36 %	N/A	N/A	< 6.5%	N/A	1.15 %
Spatial Variation	N/A	N/A	3.93 %	N/A	N/A	< 12 %	N/A	3.67 %
Linearity (ideal = 1) (LTP / LTD)	1.5 / 5.9	1.9 / 0.5	ISPP	1.51 / -0.38	1.3 / -0.3	N/A	0.96 / -0.11	1.53 / 0.47
Retention	N/A	N/A	10 ⁴ s	5,000 s	N/A	100 s	N/A	1,250 s
Endurance (Pulse number)	N/A	360	12,800	\sim 10 ⁵	N/A	4 \times 10 ³	N/A	2\times10⁶
Simulation Accuracy [%]	84.6	N/A	91.1	91.7	86.82	87.3	93.26	93.02
CMOS Compatibility	No	No	Yes	Yes	Yes	No	No	Yes

ISPP (Incremental Step Pulse Programming) is not identical pulse

Table R1. Comparison with the various synaptic transistors for neuromorphic computing.

Comment #4:

In the experimental section, the authors mention that the simulation of the GIFET based crossbar array was conducted based on “NeuroSim+”. More details about how to model crossbar array and perform network simulation should be provided in the article.

Response: We would like to thank the reviewer for suggesting this valuable comment. To evaluate the performance of GIFET as a neuromorphic device, we utilized the default analog embedded non-volatile memory (eNVM) crossbar array configuration of “NeuroSim+” as a crossbar array setting. The simulation consists of a synaptic array part and a peripheral circuit. The peripheral circuit includes a switch matrix, crossbar WL decoder, MUX decoder, analog-to-digital read circuit, adder, and shift register. The device parameters of GIFET such as set voltage, pulse width, min/max conductance, nonlinearity, cycle-to-cycle variation, and device-to-device variation are utilized to perform the simulation (see Table R2).

To reflect the reviewer’s comment, we have revised the sentences in the manuscript on page 23, line 419.

“The simulation consists of two parts: the synaptic array and peripheral circuitry. The peripheral circuit includes a switch matrix, crossbar WL decoder, MUX decoder, analog-to-digital read circuit, adder, and shift register. The device parameters of GIFET such as set voltage, pulse width, min/max conductance, nonlinearity, cycle-to-cycle variation, and device-to-device variation are utilized to perform the simulation”

	[R20]	[R25]	[R26]	[R27]	[R28]	This work
Multi Layer Perceptron	400 input 100 hidden 10 output	400 input 100 hidden 10 output	400 input 100 hidden 10 output	400 input 100 hidden 10 output	400 input 100 hidden 10 output	400 input 100 hidden 10 output
Conductance states (Potentiation / Depression)	64 / 64	50 / 50	100 / 100	320 / 256	35 / 35	1,000 / 1,000
Nonlinearity (Ideal = 0) (LTP / LTD)	-0.8028 / -0.6979 (Not use identical pulses)	0.07 / -2.42	N/A (Not use identical pulses)	1.22 / -1.75	0.06 / -0.89 (Not use identical pulses)	0.96 / -0.89
G_{\max}/G_{\min}	> 10	> 100	5	> 100	> 10	> 10
Temporal Variation	YES	No	YES	YES	No	YES
Spatial Variation	YES	No	No	No	No	YES
Accuracy	91.1%	85.88%	90.6%	88%	87%	93.02%

Table R2. Comparison of the accuracy and parameters reflected in the MNIST simulation

Comment #5:

As a transistor based device, the authors should characterize the output curve to reveal more electrical properties and details about the device.

Response: We thank the reviewer for this constructive comment. Because the GIFET is a transistor-based device, we agree that it is necessary to characterize the transfer and output curves to reveal additional electrical characteristics. Therefore, we investigated the transfer and output curves through I-V characteristic measurements, and then calculated and

summarized the subthreshold swing, threshold voltage, mobility, and current ratio at 0 V in both states when electrons were filled and unfilled in the charge store layer (see Fig. R4).

Transfer curve

Figure R4a shows a typical I_D - V_G characteristic of a GIFET. We swept the gate voltage from 0 V to -6 V to fill the CSL with electrons from the gate metal (direction 1). Subsequently, when sweeping in the positive direction from -6 V to 8 V, the curve shifts to the right. Because the channel of the GIFET is a n-type silicon and the current flow is controlled by charge depletion, a positive voltage is applied to the gate to extract electrons filled in the CSL, which maintains the depletion state of the channel. As a result, the threshold voltage shifts to the right (direction 2). Therefore, GIFET has two transfer curves according to the electron-filled state (c, d) and electron-empty state (e, f) in the CSL.

Subthreshold swing calculation

Subthreshold swing of GIFET was calculated using the following equation:

$$SS = \left(\frac{\partial(\log I_D)}{\partial V_G} \right)^{-1}$$

We calculated each of the states and selected the minimum value.

Mobility calculation

Mobility was calculated using the following equation:

$$\mu = \frac{dI_D}{dV_G} \cdot \frac{L}{WC_tV_D}$$

dI_D/dV_G (transconductance, g_m) was measured through I_D - V_G characteristic measurements, L is the channel length (20 μm), W is the channel width (5 μm), C_t ($5.86 \times 10^{-4} \text{ F/m}^2$) is the total capacitance between the gate metal and the channel per unit area ($C_t = (d_{\text{oxide}}/(\epsilon_0 \times \epsilon_{\text{oxide}}) + d_{\text{CSL}}/(\epsilon_0 \times \epsilon_{\text{CSL}}) + d_{\text{BL}}/(\epsilon_0 \times \epsilon_{\text{BL}}))^{-1}$, $\epsilon_0 = 8.85 \times 10^{-12} \text{ F/m}^2$, where ϵ_{oxide} , ϵ_{CSL} , and ϵ_{BL} are the dielectric constants of SiO_2 (3.9)^{R2}, WO_x (5)^{R3}, and a-Si (13.5)^{R4}, respectively. d_{oxide} , d_{CSL} , and d_{BL} are the thicknesses of the oxide layer (25 nm), charge store layer (25 nm), blocking layer (50 nm), respectively). V_D is 1 V. The mobility in Fig. R4b was selected as the maximum value.

In Fig. R4b, the values of the subthreshold swing and mobility are worse than those of conventional SOI transistors. The high SS value of GIFET might be helpful to make rooms for analog conductance updates. We also believe that there is leakage through defective layers due to a university-level fabrication process and this leakage needs to be optimized.

Output curve

The I_D - V_D characteristics changes based on the current device resistance state, as shown in Fig. R4a. More specifically, the curve at $V_G = 0$ V in Fig. R4c starts from the state where no electrons are in CSL. As it can accommodate a large number of electrons, the initiation of an empty state can cause high current changes. When a gate voltage is applied as -1 V, the current change in Fig. R4e (filled by electrons in CSL) is much lower than that in Fig. R4c (empty in CSL). This is because the state in Fig. R4e is already filled with electrons, thus accepting additional electrons is difficult.

Similarly, the curves shown in Fig. R4f start from the state in which the electrons are mostly occupied in the CSL. The initiation of a filled state can generate large current changes by extracting electrons. However, because the state shown in Fig. R4d has already been removed, the extraction of additional electrons is difficult. Therefore, we were able to verify that the current can be controlled by both, the amount of charge in the CSL and the weight of the synaptic device.

To reflect the reviewer's comment about the electrical characteristics of the GIFET, we have added sentences in the manuscript on page 7, line 135:

“In addition, because GIFET is a transistor-based device, its electrical characteristics are measured and evaluated as a transistor (see Supplementary Fig. 2). As the number of electrons stored in the CSL by the write/erase operation varies, the I_D - V_G and I_D - V_D characteristics can be changed, which means that the weight of the synaptic device can be controlled.”

We also appended Fig. R4 and its explanations of subthreshold swing and mobility calculation into the Supplementary Information as a Supplementary Fig. 2.

Fig. R4 Transistor characteristics of the GIFET. a I_D - V_G characteristic ($V_D = 1$ V). **b** Summary table of subthreshold swing, threshold voltage, mobility, and current ratio at 0 V of GIFET. **c-f** Output curves for each state in Fig. R4a.

Comment #6:

Description error. In Supplementary Note 4, it is the XPS data in Figure S8 the authors are talking about instead of the content in Figure S6. Please check the manuscript carefully.

Response: We would like to thank the reviewer for providing the generous comment and we apologize the confusion. We have modified the description error and the XPS data is shown in Supplementary Fig. 21. We have also doublechecked the entire manuscript to avoid description error. We have revised the sentences of Supplementary Note 5 in Supplementary Information:

“To inspect the influence of Rapid Thermal Processing on the stoichiometry of the WO_x layer, XPS measurements were conducted. The 90 nm WO_x films were deposited on cleaned SiO_2 (100 nm) /Si substrate. One of the two 2 cm \times 2 cm samples was annealed with RTP at 573 K for 10 mins in O_2 atmosphere. Supplementary Fig. 21 shows the XPS data of the sample without annealing (Supplementary Fig. 21a) and with annealing (Supplementary Fig. 21b). The data was calibrated with carbon C1s XPS peak to be located at 284.8 eV⁷. As presented in Supplementary Fig. 21, the W^{5+} intensity of the WO_x film was reduced after annealing, which suggests an increase of W^{6+} . The computed stoichiometry from surface atomic ratio

increases from $WO_{2.60}$ to $WO_{2.75}$ after annealing, which implies increasing barrier height.”

Reviewer #2 (Remarks to the Author):

Response: We sincerely appreciate reviewer #2 for suggesting the meaningful reviews on our proposed Gate Injection-based Field-Effect Transistor (GIFET). To reflect the reviewer’s evaluation on the manuscript, we have revised some expressions improving clarity. Our detailed responses to the reviewer’s technical comments are provided below.

Comments to the author:

Major Comment #1:

Memories work on the non-linearity of program vs retention e.g. ms program vs 10 years (1e8s) retention is an electron injection rate modulation of $1e^{11}$. This is provided by FN tunnelling. Choosing thermionic emission is regressive as it does not have sufficient non-linearity. The improvement in a specific performance metric i.e. linearity is not unexpected but the degradation retention does not move the “main” trade-off. This is the central challenge. Achieving one specification while neglecting another key specification is not interesting. Should this be compared against volatile memory like DRAM? In well-established DRAM, linearity is excellent - much better than the present device. It works at 3.3V or even 1.8V as opposed to the present device. So without retention, the idea does not remain compelling.

Response: We thank the reviewer for providing this comment. As the reviewer mentioned, well-established DRAM secures fast write speed and excellent linearity^{R5}. However, for a capacitor-based synaptic device with a DRAM-like structure, it is difficult to create analog conductance states on a single device. Therefore, a capacitor-based device with a complex circuit configuration was implemented to create different conductance states in a unit cell. It is essential to secure analog conductance states on a single device as a prerequisite of the synapse transistor for weight update, and to maintain retention at multiple states while training and inference are completed. Capacitor-based synaptic devices cannot have a retention time of at least several tens of seconds, and this disadvantage is exacerbated when training more complex neural networks.

On the other hand, nonvolatile memories such as charge trap flash (CTF) can distinguish between states of multilevel cells, depending on how many charges are trapped in the charge trap layer, and have long retention. Therefore, research using CTF devices is being actively conducted for applications in neuromorphic computing, as well as main memory for data storage. Owing to the endurance limit caused by the operation mechanism, research has been

conducted on inference rather than online training. Our proposed GIFET will not be targeted for data storage using nonvolatile properties, and is presented as a synapse transistor for the online training of neuromorphic computing. Because the weights of synaptic transistors are frequently updated during online training, robustness in terms of endurance is required, and the data retention requirement can be partially alleviated^{R6-R9}. Synapse devices generally do not require a 10-year retention as computing devices for online training. In other words, it does not mean that retention is ignored, but rather that retention for neuromorphic computing updates must be naturally secured. Our proposed GIFET could be a good candidate as a synaptic device as shown in Table R3.

We revised the sentences to reflect the reviewer’s comment and avoid confusion in the manuscript on page 2, line 44:

“Recently, several studies utilizing conventional memory, such as DRAM and charge trap flash memory (CTF), and emerging memory devices such as PRAM and ReRAM have been reported for neuromorphic applications¹⁰⁻¹⁷.

In the case of conventional memory, the well-established DRAM secures fast write speed and excellent linearity¹⁰. However, for a capacitor-based synaptic device with a DRAM-like structure, it is difficult to create analog conductance states on a single device. Therefore, a capacitor-based device with a complex circuit configuration was implemented to create different conductance states in a unit cell. Moreover, it is essential to secure analog conductance states for weight update and to maintain retention at multiple states while training and inference are completed, but capacitor-based synaptic devices cannot have a retention time of at least several tens of seconds.

On the other hand, nonvolatile memories such as CTF can distinguish between states of multilevel cells, depending on how many charges are trapped in the charge trap layer, and have long retention¹². Therefore, research using CTF devices is being actively conducted for applications in neuromorphic computing, as well as main memory for data storage. However, owing to the endurance limit caused by the operation mechanism, research has been conducted on inference rather than online training.”

	[R29]	[R30]	[R31]	[R23]	[R32]	This work
Data Loss Rate	8.33%	17.6%	N/A	N/A	N/A	7.15%
Retention	1,000 s	1,100 s	1,000 s	100 s	600 s	1,250 s
Endurance (Pulse Number)	N/A	1×10 ³	N/A	4×10 ³	3×10 ³	2×10⁶
CMOS Compatibility	No	No	No	No	YES	YES

Table R3. Comparison of retention with other synaptic devices for neuromorphic computing.

Major Comment #2:

The author claims that the thermionic emission-based GIFET improves linearity, which leads to an accuracy of 93.03% in MNIST data classification. Similar accuracy results are also demonstrated using other synaptic devices available in the literature [1-3] but at the expense of more complex peripheral circuitry. This claim needs to be supported using a quantitative comparison between the SOTA architectures. If the impact is in the accuracy, then accuracy improvement w.r.t previous works should be shown; if the impact is in peripheral circuitry, then improvement w.r.t previous works on those lines should be shown.

Response: We would like to thank the reviewer for this helpful comment. We agree that the accuracy can be improved if the external circuit compensates. This requires an extra burden on a more complex peripheral circuitry. In addition, to objectively verify the superior performance of our proposed GIFET, it is valuable to compare its system accuracy with other synaptic transistors.

Before comparing the simulation accuracy, it is important to check what factors are considered for the MNIST simulation, such as learning algorithms and parameters reflected in the simulation. Because the simulation accuracy is significantly affected by the components, comparisons were made based on the same input data (black and white) and multi-layer perceptron (MLP). The simulation of GIFET was conducted based on “Neurosim+”. The neural network was composed of three layers to conduct supervised learning with back-propagation. The input layer had 400 nodes for 20×20 pixels of the binary MNIST image, and the hidden neuron had 100 nodes, while the output neuron had 10 nodes for the classification results, representing 0-9 digits. Stochastic gradient descent was used for weight update. To reflect the non-ideal factors of the device, nonlinearity, G_{\max}/G_{\min} ratio, cycle-to-cycle variation, device-to-device variation, and the applied pulse scheme were considered. Even though our work considers more non-ideal parameters than previous studies, the accuracy from GIFET is improved.

We evaluated the accuracy of other synapse transistors based on MLP. Although some studies seem to show better linearity than the proposed work, quantitative comparison is not possible because of the use of an incremental amplitude pulse scheme with a circuit burden for linearity. It is noted that the incremental pulse amplitude scheme is inefficient because the read pulse should be applied during training to set up proper pulse amplitude. As shown in Table R4, the GIFET shows high accuracy with fair comparison including number of conductance states, nonlinearity, on/off ratio and spatio-temporal variation. To emphasize this point in response to the reviewer’s comment, we have added sentences in the manuscript on page 19, line 346:

“Compared to other synaptic parameters of the same algorithm, the GIFET shows high accuracy with fair comparison including number of conductance states, nonlinearity, on/off

ratio and spatio-temporal variation. (see Supplementary Table 1).”

To clarify the improvements in terms of the accuracy, we have revised the Table R4 into the Supplementary Information as a Supplementary Table 1.

	[R20]	[R25]	[R26]	[R27]	[R28]	This work
Multi Layer Perceptron	400 input 100 hidden 10 output	400 input 100 hidden 10 output	400 input 100 hidden 10 output	400 input 100 hidden 10 output	400 input 100 hidden 10 output	400 input 100 hidden 10 output
Conductance states (Potentiation / Depression)	64 / 64	50 / 50	100 / 100	320 / 256	35 / 35	1,000 / 1,000
Nonlinearity (Ideal = 0) (LTP / LTD)	-0.8028 / -0.6979 (Not use identical pulses)	0.07 / -2.42	N/A (Not use identical pulses)	1.22 / -1.75	0.06 / -0.89 (Not use identical pulses)	0.96 / -0.89
G_{\max}/G_{\min}	> 10	> 100	5	> 100	> 10	> 10
Temporal Variation	YES	No	YES	YES	No	YES
Spatial Variation	YES	No	No	No	No	YES
Accuracy	91.1%	85.88%	90.6%	88%	87%	93.02%

Table R4. Comparison of the accuracy and parameters reflected in the MNIST simulation.

Major Comment #3:

The temperature variability of GIFET is unexplored in the manuscript. Please add necessary figures indicating temperature dependence on states, retention time, and endurance time. The effect of temperature variations on linearity, retention time, and writing speed will require a compensatory circuit leading to the increased complexity of the peripheral circuitry. Robustness of hardware to temperature variations during operation needs to be shown since the mechanism is now temperature-sensitive as opposed to the typical FN tunnelling.

Response: We thank the reviewer for suggesting this constructive comment. Because the GIFET is a device based on a thermionic emission mechanism, we need to demonstrate the robustness of the hardware against temperature variations during operation. We have investigated the effect of temperature variations on the linearity, retention time, and endurance at different temperatures, to reflect the reviewer’s comment.

First, Fig. R5 shows the linear characteristics of GIFET. We confirmed the robustness of the linearity over a temperature change from 298 K to 393 K for each individual pulse duration, regardless of the write speed during 1,000 potentiation (4 V) and 1,000 depression (-4 V) operations (see Fig. R5a-d).

Fig. R5 Linearity characteristics in different temperatures. Linearity characteristics on the single device were observed with 1,000 potentiation (4 V) and 1,000 depression (-4 V) gate pulses under different temperature from 298 K to 393 K at each pulse duration **a** 200 μs , **b** 300 μs , **c** 400 μs , and **d** 500 μs .

Second, Fig. R6 shows the retention characteristics of GIFET. To investigate the retention, the drain current was measured every 1 s at 1 V for 100 s after the depression pulse train (-5 V, 500 μs) at various temperatures from 298 K to 393 K (see Fig. R6a). As shown in Fig. R6b, the small data loss was observed within 2% over the temperature range of 298 K to 373 K. The data loss at 393 K was approximately 6%, which was higher than at other temperatures, but it still showed long-term plasticity characteristics. It is more likely that the electrons will jump the barrier height easily at high temperatures because they can have more kinetic energy. Considering the normal operating temperature ($< 100\text{ }^\circ\text{C}$), the proposed GIFET can achieve data holding ability for online training.

Fig. R6 Retention characteristics of GIFET at various temperatures. **a** Current

measurement after the erase operation in various temperatures from 298 K to 393 K. Depression pulse (-5 V, 500 μ s) was applied to the gate 2,000 times, and the drain current was measured every 1 s with 1 V for 100 s. **b** Percentage of drain current change with temperature. The variation of the current value in different temperatures was measured at Fig. R6a. ΔI is the difference between the initial (I_0) and final current.

Finally, Fig. R7 and R8 show the endurance characteristics of GIFET. To investigate the robustness of the hardware, the endurance was measured under the same pulse conditions over a temperature change from 323 K to 393 K. A pulse train consists of ten consecutive potentiation pulses with an amplitude of 6 V and width of 500 μ s, followed by ten consecutive depression pulses with an amplitude of -6 V and width of 500 μ s. As presented in Fig. R7, the device operates stably by holding its high-level and low-level states, without severe degradation after 10^5 switching cycles (2×10^6 pulses in total). It means that GIFET achieved reliable endurance characteristics for highly frequent updates during online learning.

Fig. R7 Endurance characteristics of the GIFET in different temperatures. Over 10^5 switching cycles (2×10^6 pulses) were measured. Each switching cycle is composed of 10 potentiation pulses with 6 V, 500 μ s and 10 depression pulses with -6 V, 500 μ s at **a** 323 K, **b** 353 K, **c** 373 K, and **d** 393 K.

Fig. R8 Endurance variation of the GIFET in different temperatures after 10^5 switching cycle (2×10^6 pulses). The whisker represents minimum and maximum values and the box range indicates drain current values in the 25 to 70% percentile.

We have added the figures of Fig. R5d, Fig. R6a, and Fig. R8 in the Fig. 4 in the manuscript:

Page 15, line 278:

“

Fig. 4 The Robustness of the GIFET to temperature variations. Linearity variation in various temperatures from 298 K to 393 K with **a** 1,000 potentiation pulses (4 V, 500 μ s) and **b** 1,000 depression pulses (-4 V, 500 μ s). **c** Retention characteristics measurement in various temperatures from 298 K to 393 K. Depression pulses (-5 V, 500 μ s) was applied to the gate 2,000 times, and the drain current was measured every 1 s with 1 V for 100 s. **d** Comparison of endurance characteristic in different temperatures after 10^5 switching cycle (2×10^6 pulses). The whisker represents minimum and maximum values and the box range indicates drain current values in the 25 to 70% percentile.”

We added the figures of the Fig. R5a-c, Fig. R6b, and Fig. R7 as Supplementary Fig. 16-18 in Supplementary Information. We have also appended subsection “The Robustness of the GIFET to temperature variations” in the manuscript to reflect the reviewer’s comment.

“The Robustness of the GIFET to temperature variations

The revised version of Fig. 4a and b present the linear characteristics of the GIFET. We confirmed the robustness of the linearity over a temperature change from 298 K to 393 K during 1,000 potentiation (4 V, 500 μ s) and 1,000 depression (-4 V, 500 μ s) operations. As shown in Supplementary Fig. 16, the linearity of conductance update is stable at all

temperatures without severe degradation.

To investigate the retention, the drain current was measured every 1 s at 1 V for 100 s after the depression pulse train (-5 V, 500 μ s) at various temperatures from 298 K to 393 K (see Fig. 4c). As shown in Supplementary Fig. 17, the small data loss was observed within 2% over the temperature range of 298 K to 373 K. The data loss at 393 K was approximately 6%, which was higher than at other temperatures, but it still showed long-term plasticity characteristics. It is more likely that the electrons will jump the barrier height easily at high temperatures because they can have more kinetic energy. Considering the normal operating temperature (< 100 °C), the proposed GIFET can hold data for online training.

Figure 4d and Supplementary Fig. 18 shows the endurance characteristics of GIFET. To investigate the robustness of the hardware, the endurance was measured under the same pulse conditions over a temperature change from 323 K to 393 K. The pulse train consists of ten consecutive potentiation pulses with an amplitude of 6 V and width of 500 μ s, followed by ten consecutive depression pulses with an amplitude of -6 V and width of 500 μ s. As presented in Fig. 4d and Supplementary Fig. 18, the device operates stably by holding its high-level and low-level states, without severe degradation after 10^5 switching cycles (2×10^6 pulses in total). This indicates reliable endurance characteristics for highly frequent updates during online learning.”

Major Comment #4:

The authors show a dynamic range of 10x with 1000 pulses used to achieve that. FN tunnelling-based transistor memories with a much higher dynamic range can always reduce their dynamic range to improve linearity in the reduced dynamic range. These devices have much better retention [4]. How do authors motivate this change of writing mechanism for improved linearity in this case?

Response: We appreciate the reviewer’s comment. As the reviewer mentioned, FN tunnelling-based transistors can improve linearity by using a reduced dynamic range^{R10}. However, the work by Bhatt the reviewer mentioned does not show linear conductance update with consecutive pulses and the linearity is described indirectly as a function of the threshold voltage shift. In other studies, tunnelling-based transistors achieved good linearity by utilizing either incremental pulse amplitude or incremental pulse width train^{R11,R12}. This may cause complex circuitry and a long training time cost owing to the read process immediately after every single write/erase pulse.

A synaptic device for online training is required to have various characteristics such as linearity, endurance, and multi-conductance states. The endurance of FN tunnelling-based transistors is particularly disadvantageous because of the oxide damage caused by tunnelling. Because the weights of synaptic transistors are frequently updated during online training, robustness in terms of endurance is required, and the data retention requirement can be

partially alleviated^{R6-R9}. In other words, this does not mean that retention is ignored, but rather that retention for neuromorphic computing updates will be naturally secured under updates during online training.

One of the aims of our study is to demonstrate a gate injection-based field-effect synapse transistor to overcome the limitations of endurance owing to the tunnelling-based mechanism. Therefore, we were motivated to deviate from the existing devices and improve the linearity and endurance of the conductance update as a synapse transistor for online training of neuromorphic computing. We also believe that further improvement of retention while maintaining robust endurance is the direction of our future research for various applications.

Major Comment #5:

The main manuscript should include a comparison with other synaptic devices, clearly highlighting improved factors/advantages. (Critical parameters to be included in the comparison are Area, Power, Linearity, Programming time, Retention time, Accuracy benefits, temperature dependence, and variability.)

Response: We would like to express our gratitude to the reviewer for this constructive suggestion. Table R5 shows the comparison of the properties of GIFET versus other types of synaptic transistors in terms of the number of conductance states, area, power consumption, on/off ratio, operating voltage, programming time, spatio-temporal variation, linearity, retention, endurance, simulation accuracy and CMOS compatibility. It is noted that the area can be reduced as the technology node used for the fabrication improves. As shown in the table, GIFET has superior properties than other synaptic transistors.

To reflect the reviewer's comment, we have revised the sentences in the manuscript on page 20, line 360:

“GIFET shows superior properties such as number of conductance states, area, power consumption, on/off ratio, operating voltage, programming time, spatio-temporal variation, linearity, retention, endurance, simulation accuracy and CMOS compatibility (see Table 1).”

We have added the Table R5 as Table 1 in the manuscript on page 17, line 325.

	[R18]	[R19]	[R20]	[R21]	[R22]	[R23]	[R24]	This work
The Number of Conductance States (LTP / LTD)	100 / 100	60 / 60	64 / 64	2^{13} / 2^{13}	100 / 50	50 / 50	100 / 100	1,000 / 1,000
Area [μm^2]	0.52	3	15,000	350	N/A	N/A	2,000	100
Power Consumption [fJ]	< 400	30	\approx 60,000	< 20	N/A	\approx 160	N/A	< 50
On/off ratio	\approx 3	\approx 2	> 10	\approx 10^3	> 10	\approx 2	\approx 9	> 10
Program Voltage [V]	2	1.2	2.7 ~ 4.3	4	3.5	2.5	3	1.8
Program Time [ms]	100	100	10	10	10	10	100	0.3
Temporal Variation	N/A	N/A	2.36%	N/A	N/A	< 6.5%	N/A	1.15%
Spatial Variation	N/A	N/A	3.93%	N/A	N/A	< 12%	N/A	3.67%
Linearity (ideal = 1) (LTP / LTD)	1.5 / 5.9	1.9 / 0.5	ISPP	1.51 / -0.38	1.3 / -0.3	N/A	0.96 / -0.11	1.53 / 0.47
Retention	N/A	N/A	10^4 s	5,000 s	N/A	100 s	N/A	1,250 s
Endurance (Pulse number)	N/A	360	12,800	$\sim 10^5$	N/A	4×10^3	N/A	2×10^6
Simulation Accuracy	84.6%	N/A	91.1%	91.7%	86.82%	87.3%	93.26%	93.02%
CMOS Compatibility	No	No	Yes	Yes	Yes	No	No	Yes

ISPP (Incremental Step Pulse Programming) is not identical pulse

Table R5. Comparison with the various synaptic transistors for neuromorphic computing.

Major Comment #6:

At its present state, the synapse area is $5\mu\text{m} * 20\mu\text{m}$, which is very large compared to SOTA synaptic devices. The gate stack is 75 nm thick. The device should not be scalable below 75 nm in the horizontal direction without incurring severe fringing fields, interFG interference, and short channel effects. The smaller size is an essential parameter for achieving a higher packing density of synapses in neural circuits. Is there any possibility of scaling the device, and will that affect the conduction mechanism and linearity? Without this analysis, it is difficult to assess the relevance of this device.

Response: We are grateful to the reviewer for providing the insightful comment. The measured synapse area of the device was $5\mu\text{m} \times 20\mu\text{m}$, which is fairly large compared with that of SOTA synaptic devices. Scaling down to the nanoscale is required for a higher packing density, and we agree that additional investigations need to be performed to see the

effect of fringing field, interFG interference and short channel effect. We measured the linearity of conductance update as a function of device size, and the linearity was not affected by the device scaling. We agree that nanometer-size device needs to be examined. Unfortunately, patterning sub-micrometer-scale devices is difficult in this research due to the fact that GIFET was fabricated using a university-level process rather than a CMOS-based foundry process. we focused on the concept of GIFET in this manuscript, and we will study the consequence of reduced size as a future study as the reviewer mentioned.

To analyze the conduction mechanism and linearity according to the device scaling, as in the currently available experiments, we additionally measured the LTP-LTD characteristics of various widths and lengths of the synapse area at several temperatures. Figures R9a and b show a comparison of the nonlinearity of potentiation and depression, respectively, at different temperatures and synapse area sizes. Potentiation (4 V, 500 μ s) and depression (-4 V, 500 μ s) pulse trains were applied to the gate 1000 times each, and the nonlinearity values were calculated. Regardless of temperature and synapse area, there was no significant difference in nonlinearity values for both potentiation and depression cases, which means that the conduction mechanism of the GIFET is insensitive to size and can maintain its linearity. Even if the ratio of the width to length of the synapse area is the same, the current level is different (see Fig. R9c). Because the GIFET was designed in the structure of an n-n-n junctionless transistor with the same channel length of the mesa pattern and without defining the source and drain, the effective current-flowing channel width and length could be different (see inset of Fig. R9c).

In this study, we focused on the possibility of linearly updating synaptic weights by injecting and extracting directly between the gate metal and the charge store layer. We expect to minimize uncertainties and optimize device characteristics by changing the device structure by defining the source and drain areas or designing the gate to finFET in the future.

Fig. R9 Linearity and I_D - V_G characteristics of GIFET depending on the width and length of synapse area. Comparison of **a** potentiation and **b** depression nonlinearity values in various widths and lengths of synapse area at several temperatures. Potentiation (4 V, 500 μ s) and depression (-4 V, 500 μ s) pulse trains were applied 1,000 times. **c** I_D - V_G characteristic measurement in different widths and lengths of synapse area. Inset: an optical microscope image of GIFET.

Minor Comment #1:

The author has not shown the effect of device-to-device variability on the Linearity of the device. Please provide a figure similar to 3c and 3d indicating device-to-device variability. Also, consider device-to-device variability in the NeuroSim+ simulation to calculate the accuracy of MNIST classification.

Response: We agree that it is necessary to provide spatial variation for the synaptic devices. Therefore, we confirmed the device-to-device variation for LTP-LTD operation. The spatial variation was defined as the standard deviation of the nonlinearity baseline in the NeuroSim+ simulation. The following equation was used to calculate the nonlinearity of the GIFET in the simulation:

$$G_{LTP} = B \left(1 - e^{\left(\frac{P}{A}\right)} \right) + G_{min}$$

$$G_{LTD} = B \left(1 - e^{\left(\frac{P-P_{max}}{A}\right)} \right) + G_{max}$$

$$B = (G_{max} - G_{min}) / \left(1 - e^{\frac{-P_{max}}{A}} \right)$$

where G_{LTP} and G_{LTD} are the conductance of LTP and LTD, respectively. G_{max} , G_{min} , and P_{max} were directly extracted from the experimental data. A is a parameter that controls the nonlinear behavior of the weight update. This value is close to 0, which represents ideal linearity^{R13}.

Fig. R10 Effect of device-to-device variation on the LTP/LTD. **a, b** Normalized LTP-LTD characteristics. **a** Potentiation device variation under 1,000 potentiation pulses (4.5 V / 500 μ s). **b** Depression device variation under 1,000 depression pulses (-4.5 V / 500 μ s). **c** Box plot of LTP-LTD characteristic and device-to-device variation based on the measured data of Fig. R10a and b. **d** Simulation accuracy of the MNIST image classification reflecting nonlinearity values with/without device-to-device variation of the LTP-LTD.

As shown in Fig. R10a, b, and c, we measured the device-to-device variation in LTP/LTD. The nonlinearity was also extracted for each device using the equation in Supplementary Note 2 (same as above equation). The standard deviation of the nonlinearity (σ_{NL}) was calculated as 0.60 / 0.44 during potentiation/depression in Fig R10a and b. The device-to-device variation of LTP/LTD was also investigated as 20.12% / 24.49% in Fig. R10c. This is fairly larger than the cycle-to-cycle variation (\sim 2.55%) due to the inherent variation of forming the 20 nm Si layer from SOI wafer using thermal oxidation and wet etching process. Besides, the overall use of university-level process could have increased the device-to-device variation.

Based on these measured data, we conducted an MNIST classification simulation based on “NeuroSim+”, including device-to-device variability. Figure R10d presents the accuracy of

MNIST classification for each epoch. We also obtained accuracy of approximately 93.02% during 5,000 epochs. The GIFET showed an accuracy of 93.03% without considering the device-to-device variation, and with difference of ~0.01%. We confirmed that device-to-device variation did not degrade the accuracy within the characteristics of GIFET.

To reflect the reviewer’s comment, we revised the figures and sentences in the manuscript:

Page 19, line 342: “As observed, the GIFET based artificial neural network obtained accuracy of approximately 93.02% during 5,000 epochs, which is almost equivalent to that of an ideal linearity device, 93.45%.”

Page 13, line 241: “In addition, the repeated LTP-LTD characteristics on the single device were observed with 1,000 potentiation (500 μ s, 1.4 V) – 1,000 depression (500 μ s, -2.5 V) gate pulse trains, as displayed in Fig. 3c. The LTP-LTD characteristics of the device in Fig. 3d presented low variation of 2.55% (σ/μ at the 1,000th pulse). The device-to-device variation and standard deviation of nonlinearity based on the LTP-LTD characteristics of several devices are also calculated in Supplementary Fig. 13. The nonlinearity during potentiation/depression in Supplementary Fig. 13b and c was fitted using method from Supplementary Note 2.”

To clarify this, we have added the Fig. R11 as Supplementary Fig. 13 and the sentences in the Supplementary Note 2:

Fig. R11 Device-to-device variation of the GIFET on the LTP/LTD. a Box plot of LTP-LTD characteristics and device-to-device variation for 10 devices (potentiation: 4.5 V / 500 μ s, depression: -4.5 V / 500 μ s). **b, c** Normalized LTP-LTD characteristics extracted from Fig. R11a. Nonlinearity value was fitted using method from Supplementary Note 2.”

“Supplementary Note 2. Nonlinearity in the simulation.

The spatial variation was defined as the standard deviation of the nonlinearity baseline in the NeuroSim+ simulation. To reflect the device-to-device of the GIFET in the MNIST classification simulation, the following equation was used to calculate the nonlinearity of the GIFET in the simulation:

$$G_{LTP} = B \left(1 - e^{\left(\frac{P}{A}\right)} \right) + G_{min}$$

$$G_{LTD} = B \left(1 - e^{\left(\frac{P-P_{max}}{A}\right)} \right) + G_{max}$$

$$B = (G_{max} - G_{min}) / \left(1 - e^{\frac{-P_{max}}{A}} \right)$$

where G_{LTP} and G_{LTD} are the conductance of LTP and LTD, respectively. G_{max} , G_{min} , and P_{max} were extracted directly from Supplementary Fig. 13b, c. A is a parameter that controls the nonlinear behavior of the weight update. This value is close to zero, which represents the ideal linear conductance change⁶.”

Minor Comment #2:

The authors claim that the use of two-terminal devices as synapses suffers from device-to-device variation and operation stochasticity. However, the proposed GIFET also suffers from device-to-device variability, as shown in fig. 3b. A quantitative comparison is necessary to indicate reduced variability.

Response: We appreciate the reviewer for providing the helpful comment. We agree that a quantitative comparison with memristors is necessary to indicate low variability and highlight our strengths. To compare the device-to-device variability, the variation was calculated as σ/μ of the set voltage and reset voltage. Set voltage refers to the voltage that changes from a high-resistance state (HRS) to a low-resistance state (LRS), and reset voltage is the voltage that changes from LRS to HRS. As shown in Table R6, GIFET exhibits the advantage of low variability.

In the case of memristors, which are two-terminal devices, the conductance changes are described as randomly formed filament during set/reset. This stochastic behavior of ion movement causes unreliable variations. Therefore, studies have focused on improving the operational reliability (device-to-device variation) of memristor as shown in Ref 32,33,36. However, the proposed GIFET has shown high uniformity intrinsically because it uses an electron population for conductance updates instead of stochastic movement of ions.

We have added sentences to clarify the intention and avoid confusion in the manuscript on page 3, line 64:

“However, their device variation caused by randomly formed filament during set/reset. This stochastic behavior of ion movement causes unreliable variation and it has been the significant bottleneck for the successful application as a computing device^{4,19}”

	[R32]	[R33]	[R34]	[R35]	[R36]	This work
Set Voltage Variation (σ/μ)	~5%	5.7%	17%	~19.5%	8.76%	5.16%
Reset Voltage Variation (σ/μ)	N/A	N/A	21%	~16.8%	N/A	3.67%

Table R6. Comparison of device-to-device variation with two-terminal memristors

Minor Comment #3:

The author is requested to add plots indicating the relationship between write pulse amplitude and conductance change per pulse. Also, add a similar graph showing a relationship between the write pulse duration and conductance change per pulse.

Response: We would like to thank the reviewer for suggesting the valuable comment. The relationship between pulse amplitude/duration and conductance change per pulse is important, because it is related to the stable control of the pulse scheme. We have conducted additional experiments to reflect the reviewer's comment. First, Fig. R12 shows the relationship between the pulse amplitude and current change at different pulse durations. Because the current change is small in short pulse durations, a larger voltage than usual was used to clearly observe the current change. As shown in Fig. R12a-c, linear conductance updates were observed in all cases. As expected, the degree of update is smaller for shorter durations or lower voltages (see Fig. R12d).

Fig. R12 Relationship between pulse amplitude and current change. The current at different depression pulse amplitude from -9 to -10 V at each pulse duration **a** 500 μs , **b** 400 μs , **c** 300 μs . **d** Current change per 100 pulses extracted from Fig. R12a–c.

Second, Fig. R13 shows the relationship between the pulse duration and current change at a fixed pulse amplitude. As shown in Fig. R13a, the conductance update was performed until saturation with different pulse durations, ranging from 100 μs to 700 μs . The longer pulse duration increases ΔI and the current is saturated quickly as shown in Fig. R13b.

Fig. R13 Relationship between pulse duration and current change. **a** Drain current at different depression pulse duration from 100 to 700 μ s at fixed -10 V pulse amplitude. **b** Current change extracted from Fig. R13a.

To reflect the reviewer’s suggestion, we added the Fig. R12 and 13 as the Supplementary Fig. 8 and 9 into the Supplementary Information. We have also added the following sentences in the manuscript on page 11, line 208:

“Next, the relationship between operation pulse amplitude/duration and conductance change per pulse was investigated as shown in Supplementary Fig. 8. It shows the relationship between the pulse amplitude and current change at different pulse durations. GIFET can control the linear conductance update with small current change per pulse in various pulse scheme (see Supplementary Fig. 9-11). In other words, the GIFET shows stable characteristics for controlling the pulse scheme for neuromorphic computing applications, and has the advantage of being customizable to fit the needs of other applications.”

Minor Comment #4:

How does lower retention time affect the training? What is the plan for inference which requires long-term non-volatile weight storage? (As the conduction mechanism is changed to thermionic for which the barrier height between CSL and Blocking layer is decreased)

Response: We appreciate the reviewer for providing the comment. Retention must be guaranteed for memory applications, and the synaptic transistor should be improved for as long as possible to utilize various applications. Retention during parallel weight updates must be ensured for neuromorphic computing. However, because the weights of the synaptic transistors are frequently updated for online training, retention can be partially alleviated^{R6-R9}. This means that the retention time must be longer than the time from the update to the next

update, or more than the time to infer from the weights for reliable online training.

Several recent studies investigated the effects of training and retention using the MNIST dataset. The required retention of the device was related to the time of each training cycle, which was affected by the individual code and algorithm. In the case of a 2 ms training cycle, the effect on the training result was negligible for a minimum time constant of 20 s on the MNIST dataset^{R14}. The time constant is defined as the time that conductance value decreases to 36.8% (e^{-1}) from the initial state and remains at this level. In the case of 200 ns training cycle, test error of simulation also was negligible for a minimum time constant of 0.2 s on the MNIST dataset^{R5}. A decrease of 7.15% (5 nS) was observed for 1,200 s and time constant was calculated as 6,176 s, by converting to a proportional formula in our proposed GIFET. It means that retention of GIFET for online training was sufficiently secured, and the accuracy loss due to retention was not significant.

GIFET was targeted as a computing device for online training rather than a storage device. In computing applications that require long-term weight storage, the computed data from GIFET could be transferred in conventional memory and reload the data to the GIFET. However, this does not occur frequently during computing process.

Minor Comment #5:

Extend plot of I_d vs. pulse number till I_d saturates during the writing operation and erasing operation. It shall provide a maximum/minimum capacity of the charge storage layer and maximum dynamic range.

Response: We thank the reviewer for this helpful suggestion. To investigate the capacity of the charge store layer and maximum dynamic range of GIFET, we further measured the LTP-LTD characteristics until the drain current was saturated.

As shown in Fig. R15, the drain current was measured for 10,000 pulses for write and erase operations. In the case of write operation (5 V, 500 μ s), the current increases linearly until ~1,200 pulses and starts to saturate at ~6,000 pulses (see Fig. R15a). Comparably, in the case of erase operation (-5 V, 500 μ s), the current decreases linearly until ~1,500 pulses and starts to saturate at ~5,000 pulses (see Fig. R15b). The dynamic range from the initial state to the saturation region exceeds ~200 for both operations. However, since the degree to which GIFET is updated per pulse depends on the pulse amplitude and duration, the linear update region and saturation point can be changed flexibly (please refer to our earlier analysis: comment #3 from reviewer #1 and minor comment #3 from reviewer #3).

Fig. R15. Extended plot of I_D versus pulse number. **a** Extended I_D versus pulse number plot for write operation (5 V, 500 μ s). Drain current linearly increases until approximately 1,200 pulses and starts to saturate from 6,000 pulses. **b** Extended I_D versus pulse number plot for erase operation (-5 V, 500 μ s). Drain current linearly decreases until approximately 1,500 pulses and starts to saturate from 5,000 pulses.

To reflect the reviewer's suggestion and supplement the additional experiments, we added Fig. R15 as Supplementary Fig. 11 into Supplementary Information.

Minor Comment #6:

Check figure S1b, the band diagram.

Response: We appreciate the comment. Band diagram of the write operation is shown in Fig. R16a (from figure S1b that the reviewer mentioned) and R16c. We doublechecked the band diagram of GIFET by using the TCAD tool in case of inserting SiO₂ as the interfacial layer (ref. Fig. R16b). As shown in Fig. R16c, electrons are extracted from the CSL to the blocking layer to perform the write operation. Therefore, a positive voltage is applied to the gate to extract electrons during the write operation.

We revised sentences in Supplementary Fig. 4 to reflect the reviewer’s comment and to avoid the confusion:

“Band diagram of changed conduction mechanism by interfacial layer insertion during the write operation (electrons migrated from CSL to blocking layer)”

Fig. R16 Band diagram simulation of the GIFET using TCAD. a Band diagram of changed conduction mechanism by interfacial layer insertion during the write operation in Supplementary Fig. 2b. **b** Structure of the GIFET inserted interfacial layer. **c** Band diagram based on TCAD tool from CSL to blocking layer during the write operation.

Minor Comment #7:

Page 3, line 64: correct the spelling to Fowler-Nordheim tunnelling.

Response: We would like to appreciate the reviewer for providing the meticulous comment. We have changed all relevant parts of the manuscript to reflect the reviewer’s comment from ‘tunneling’ to ‘tunnelling’ in the revised manuscript:

Page 4, line 79: “The conventional three-terminal floating gate-based flash memory shows high nonlinearity in weight updates^{12,21,29,30} due to the Folwer-Nordheim (F-N) tunnelling, a vital function of the electric field changed by electrons stored charge state^{29,31}”

Page 5, line 95: “The dependence of current density through tunnelling oxide on the current

floating gate charge state triggers the nonlinearity of conventional flash memory due to its primary update mechanism, F-N tunnelling²⁹⁻³¹”

Page 9, line 167: “Accordingly, the conduction mechanism between CSL and blocking layer converted from thermionic emission to field emission or trap assisted tunnelling through SiO_x layer (see Supplementary Fig. 4b). Supplementary Fig. 4c shows the Arrhenius plot of $\ln(I/T^2)$ versus q/kT for the device with the interfacial layer. The zero slope of the graph shows that the disappearance of the Schottky barrier and the linear relationship between $\ln(I/V^2)$ and $1/V$ on the device with the interfacial layer at room temperature in Supplementary Fig. 4d implies the changed conduction mechanism is F-N tunnelling⁴¹”

Reviewer #3 (Remarks to the Author):

The manuscript reports a gate injection based field-effect transistor with superb synapse performance in artificial neural networks. The transistor shows high linearity on long-term plasticity with the mechanism of thermionic emission. The work is valuable to realizing CMOS-compatible neuromorphic computing. Before publication there are several problems that need to be addressed by the authors.

Response: We appreciate reviewer #3 for suggesting valuable comments on our work and for recognizing the novelty of our proposed gate injection-based field-effect transistor (GIFET). To reflect the reviewer’s evaluation on the manuscript, we have revised grammar and expression improving clarity. Our detailed responses to the reviewer’s technical comments are provided below

Comment #1:

What are the exact values of the energy levels in Fig. 1b? The authors need to provide experimental evidence or references for the band diagram.

Response: We appreciate the suggestion. We selected the charge store layer material (4.92 eV)^{R15} with a larger electron affinity than the blocking layer (3.93 eV)^{R16} to store enough charge for a high current on/off ratio and to hold electrons for long retention. We refer to the exact values of the charge store layer and blocking layers below the references.

R15. Liu, X., Zheng, H., Li, Y. & Zhang, W. Factors on the separation of photogenerated charges and the charge dynamics in oxide/ZnFe₂O₄ composites. *J. Mater. Chem. c* **1**, 329–337 (2013).

R16. Matsuura, H., Okuno, T., Okushi, H. & Tanaka, K. Electrical properties of n-amorphous/p- crystalline silicon heterojunctions. *J. Appl. Phys.* **55**, 1012–1019 (1984).

To clarify the details above in response to the reviewer's comments, we have appended the exact values and references in the manuscript:

Page 6, line 109: "As the current density through the thermionic emission depends on the barrier height between each layer, the band diagram of the GIFET was designed as shown in Fig. 1b. The barrier height between the charge store layer (CSL) and blocking layer should be high enough to store charge for a high current on/off ratio and to hold electrons for long retention. Therefore, we selected a CSL material ($\chi_{CSL} = 4.92$ eV)³⁹ with a larger electron affinity than the blocking layer ($\chi_{BL} = 3.93$ eV)⁴⁰."

Comment #2:

Does the transistor work in a depletion mode? If so, how is the depletion region in the channel formed?

Response: We would like to thank the reviewer for the helpful comment. The device does work in a depletion mode. Our device was designed in the form of an n-n-n junctionless transistor, and detailed 2D schematic diagrams of the operation processes were prepared for a more accurate understanding, as shown in Fig. R17.

For the erase process, a depletion layer was formed under the gate oxide in accordance with the electrons stored in the charge store layer (CSL) by applying a negative voltage to the gate stack. Because the electrons stored in the CSL are negative, the electrons in the n-type channel will be removed due to repulsive force, and the depletion region will be created in the channel.

For the write process, a positive bias was applied to the gate while the source and drain were grounded. Negatively charged electrons on the WO_x layer were extracted to the gate metal due to the electric field. Subsequently, it results in shrinking of the depletion region in the channel.

To read the stored weight of the device, the gate is grounded, and the read voltage is applied to the drain. The synaptic weight is obtained by measuring the resistance between the source and drain. Therefore, by controlling a portion of the depletion region in the channel, the synaptic weight can be stored.

Fig. R17 Detailed operation principle of the GIFET in 2D schematic diagrams. a Initial state. **b-d** Write operation sequence; **b** electron extraction, **c** reduction of depletion region, and **d** expansion of channel. **e-g** Erase operation sequence; **e** electron injection, **f** expansion of depletion region, and **g** reduction of depletion region.

To reflect the reviewer’s comment and to provide a better understanding of the operating principles of GIFET, we added the following sentence in the revised manuscript on page 7, line 134:

“More detailed operating principles are described in the Supplementary Fig. 1.”

We also appended Fig R17 and its explanations into Supplementary Information as Supplementary Fig. 1.

Comment #3:

Why the absolute value of writing voltage is not equal to that of the erasing voltage? It is interesting that the absolute value of writing voltage is higher in Fig. 2c and lower in Fig. 3b. Could the authors clarify the reason for choosing the writing and erasing voltage?

Response: We appreciate the meticulous comment on our work. Our proposed GIFET was designed to have a difference in barrier height for the write and erase operations. During the write operation in Fig. R18a, electrons are extracted from the CSL to the gate through the blocking layer. The barrier height formed by the CSL and blocking layer plays an important role in the conduction mechanism of the write operation. However, during the erase operation in Fig. R18b, electrons are injected from the gate to the CSL through the blocking layer. In this case, the barrier height formed by the gate metal and the blocking layer plays an important role in the erase operation.

Fig. R18 Schematic of write/erase operation from energy band diagram. a Write operation. Electrons are extracted from the charge store layer to the gate metal. **b** Erase operation. Electrons are injected from the gate metal to the charge store layer. These two operations are dominantly based on thermionic emission mechanism.

To experimentally obtain the barrier height for each operation, we obtained the Arrhenius plot by utilizing the same stack as in Fig. 2a (identical stacks are described in Fig. R19a and c). A linear relationship between $\ln(I/T^2)$ and q/kT was observed through the Arrhenius plot, and we confirmed that the barrier heights are different between the write and erase operations as shown in Fig. R19b and d. Consequently, because the barrier height is structurally different for each write and erase operation, there is a difference in the absolute values of the write and erase voltages.

Fig. R19 The Arrhenius plot for relationship between $\ln(I/T^2)$ and q/kT under varied temperature from 273 K to 423 K. **a,b** Schematic of gate stack without gate oxide identical stack to Fig. 2a. The arrows indicate the direction of movement of electrons inside the stack during **a** write (applying positive bias on the gate) and **b** erase (applying negative bias on the gate) operations. **c, d** Arrhenius plot for **a** and **b**, respectively. Each activation energy is derived between CSL and the blocking layer, and between the gate metal and blocking layer, respectively.

In Fig. R20a and R20b (same as Fig. 2e and Fig. 3c in the manuscript), the voltage conditions in the manuscript (write 1.4 V / erase -2.5 V), we intended a write voltage smaller than the erase voltage for arbitrary pulse trains and stable operation because of the different barrier heights. On the other hand, the intended purpose of Fig. R20c (same as Fig. 2c in the manuscript) is to show the dependence between the linearity of the LTP-LTD and maximum on/off ratio. Therefore, the pulse condition of higher voltage in Fig. R20c (write 5 V / erase -3.3 V) than Fig. R20b (write 1.4 V / erase -2.5 V) were utilized to increase the on/off ratio.

Fig. R20 Reconfiguration of figures stated in the reviewer’s comment. a Identical figure to Fig. 2e in the manuscript. The LTP-LTD characteristics of the GIFET under arbitrary pulse trains (haphazard write/erase pulse) train consists of 500 write (500 μ s, 1.4 V) / erase (500 μ s, -2.5 V) / hold (0 V) gate pulses with a read pulse (200 μ s, 1 V) on the drain 200 μ s after each gate pulse. (red: write operation, blue: erase operation, white: hold). **b** Identical figure to Fig. 3c. The repeated LTP-LTD characteristics on a single GIFET device with 1,000 potentiation – 1,000 depression (1.4 V/-2.5 V, 500 μ s). **c** Identical figure to Fig. 2c. LTP-LTD characteristic of GIFET under 1,000 write (5 V, 500 μ s)-1,000 erase (-3.3 V, 500 μ s) consecutive pulse trains (1 V, 50 μ s read voltage and 50 μ s between each pulse were used).

In order to reflect the reviewer’s comment and experimentally verify the difference of the barrier height between the gate stack, we added Fig. R18a and Fig. R19b, d as Supplementary Fig. 3. We also appended the following sentence to refer to these supplementary data in the revised manuscript on page 9, line 162:

“More details on barrier height are in Supplementary Fig. 3”

Comment #4:

Could the authors explain the stochastic gradient descent method more clearly?

Response: We appreciate for the suggestion. Stochastic Gradient Descent (SGD) is one of gradient descent optimization algorithms for updating neural network parameters^{R17}. The gradient of the cost function for the neural network parameters was computed using a gradient descent algorithm. When using the entire training dataset, it takes a lot of time to obtain optimal parameters because the gradients for similar examples are recomputed before each parameter update. To avoid this problem, SGD randomly samples examples from the training dataset for each epoch to compute the gradients. Therefore, it is usually much faster and widely used for the training process.

To explain the stochastic gradient descent method more clearly, we have added the following sentences to the manuscript:

Page 4, line 414: “Stochastic gradient decent was used for weight update. The gradient of the cost function for the neural network parameters was computed using a stochastic gradient descent algorithm. Stochastic gradient decent randomly samples examples from the training dataset for each epoch to compute the gradients. Therefore, it is usually much faster and widely used for the training process.”⁴⁸”

Comment #5:

The authors explain the conduction mechanism by I-V characteristic measurements in the temperature range of 339 K ~ 413 K. What temperature do the authors measure the LTP-LTD characteristics at? Whether is the conduction also dominated by thermionic emission at this temperature?

Response: We would like to thank the reviewer for the insightful comment. As the reviewer commented, the LTP-LTD characteristics are measured at room temperature.

We revealed which mechanism dominates the thermionic and F-N tunnelling conduction mechanisms by measuring the structure (WO_x/a-Si:H) and modified structure (WO_x/3 nm SiO₂/a-Si:H) at the elevated temperature between 339 K and 415 K as shown in Fig. 2a, b and Supplementary Fig. 2a-c. The figures are redrawn below as Fig. R21.

Fig. R21 Reconfiguration of Fig. 2a, b and Supplementary Fig. 2a-c. **a, b** are identical to Fig. 2a, b in the manuscript, and **c, d** are identical to Supplementary Fig. 2a-b in Supplementary Information. **a** Schematic of the GIFET gate stack without gate oxide and band diagram of CSL and blocking layer during write operation. The barrier exists between CSL and blocking layer and the charge is extracted over the barrier. **b** Arrhenius plot for the relationship between the current through gate stack of GIFET without interfacial layer under varied temperature from 339 K to 415 K when applied voltage is 1 V ($\ln(I/T^2)$ versus q/kT). **c** Schematic of gate stack with interfacial layer between CSL and blocking layer and band diagram of changed conduction mechanism by interfacial layer insertion during the write operation (electrons migrated from CSL to blocking layer). **d** Arrhenius plot for relationship between $\ln(I/T^2)$ and q/kT through the blocking layer with interfacial layer under varied temperature from 345 K to 418 K.

To clarify the charge transport mechanism at room temperature, the $WO_x/a\text{-Si:H}$ structure without the SiO_2 layer was prepared as shown in Fig. R21a. The current through the gate stack of the GIFET was measured again from 273K to 423K including room temperature. The Arrhenius plot was drawn in Fig. R22. A linear relationship between $\ln(I/T^2)$ and q/kT was observed through the Arrhenius plot, indicating that the main conduction mechanism at room temperature is thermionic emission.

Fig. R22 The Arrhenius plot measured from WO_x/a-Si:H stack without the SiO₂ layer between the channel and the stack to measure the gate current through the blocking layer in the temperature range from 273 K to 423 K.

To reflect the reviewer’s comment, we revised the figure and sentences in the manuscript, as most of the electrical characteristics of the GIFET were measured at room temperature. We have replaced Fig. 2b to Fig. R22 in the manuscript on page 8, line 139, the temperature range description of the legend of Fig. 2 was modified in the manuscript:

Page 8, line 142: “The Arrhenius plot measured from WO_x/a-Si:H stack without the SiO₂ layer between the channel and the stack from 273 K to 423 K”

Page 9, line 153: “To investigate the effect of the conduction mechanism for charge transport through the blocking layer on the linearity, we observed the dependence of the current density through the blocking layer at several temperatures. To focus on the current through the blocking layer and confirm the presence of the Schottky barrier, a gate stack of the device without the gate oxide was prepared (see Fig. 2a). I-V characteristic measurements were conducted in the temperature range 273 K-423 K.”

Comment #6:

In Figure 2e the drain currents are almost the same after 500 write/erase pulses except for the W-E-W-E process. Please clearly discuss the underlying mechanisms?

Response: We appreciate the reviewer for pointing out the difference. The W-E-W-E process was performed with an additional write process at the beginning. For fair comparison, we have replaced to Fig. R23 (Fig. 2e in the manuscript).

Fig. R23 The LTP-LTD characteristics of the GIFET under arbitrary pulse (haphazard write/erase pulse) train during W-E-W-E process.

Comment #7:

For Figure 2c, e and 3f the authors have measured the LTP-LTD characteristics and endurance by using different voltages of write/erase pulses. Why do the authors not keep the voltages consistent?

Response: We thank the reviewer for providing the meticulous comment. We measured the LTP-LTD characteristics and endurance of the GIFET using different voltages, as shown in Fig. 2c, e and Fig. 3g (we carefully assume that the endurance characteristic mentioned by the reviewer refers to Fig. 3g, not Fig. 3f). These figures are also redrawn below as Fig. R24.

Fig. R24 Reconfiguration of figures stated in the reviewer's comment. **a** Identical figure to Fig. 2c in the manuscript. LTP-LTD characteristic of GIFET under 1,000 write (5 V, 500 μ s)-1,000 erase (-3.3 V, 500 μ s) consecutive pulse trains (1 V, 50 μ s read voltage and 50 μ s between each pulse were used). **b** Identical figure to Fig. 2e. LTP-LTD characteristics of the GIFET under arbitrary pulse trains (haphazard write/erase pulse) train consists of 500 write (500 μ s, 1.4 V) / erase (500 μ s, -2.5 V) / hold (0 V) gate pulses with a read pulse (200 μ s, 1 V) on the drain 200 μ s after each gate pulse. (red: write operation, blue: erase operation, white: hold). **c** Identical figure to Fig. 3g. Endurance of GIFET over 10^5 switching cycle (2×10^6 pulses). Each switching cycle is composed of 10 write pulses with 6 V, 500 μ s and 10 erase pulses with -6 V, 500 μ s.

First, the intended purpose of Fig. R24a (same as Fig. 2c) is to observe how the linearity of the LTP-LTD is maintained while increasing the on/off ratio. Therefore, the pulse condition of higher voltage in Fig. R24a (write 5 V / erase -3.3 V) than Fig. R24b (same as Fig. 2e, write 1.4 V / erase -2.5 V) was needed to increase the on/off ratio. Second, Fig. R24b shows a linear conductance update with arbitrary pulse trains. As reviewer #1 noted, it is important for synaptic devices to maintain nonvolatile states at different intermediate conductance levels for neuromorphic computing application. We required a smaller voltage in Fig. R24b than in Fig. R24a to verify the intermediate states. Therefore, we changed the voltages for Fig. R24a and R24b.

Figure R24c (same as Fig. 3g) shows the endurance characteristics of the GIFET. Reliable

neuromorphic computing applications are crucial. Because there was a limitation to test endurance in normal condition, the applied voltage of the endurance test was increased to create severe circumstance to prove robust reliability.

We added sentences to reflect the reviewer's suggestion in the manuscript:

Page 11, line 198: "Figure 2e presents the linear conductance update with arbitrary pulse trains. Pulses consisting of 500 write pulses (500 μ s, 1.4 V), 500 hold pulses (0V) and 500 erase pulses (500 μ s, -2.5 V), and 500 hold (0 V) were applied to the gate. For the read process, a read pulse (200 μ s, 1 V) is applied to the drain terminal. As shown in Fig. 2e, during the hold process, current change has not been observed, which means that the data is well preserved. It is important for synaptic devices to maintain nonvolatile states at different intermediate conductance levels for neuromorphic computing applications."

Comment #8:

The devices with lightly- and highly-deteriorated linearity would exhibit degraded training results of artificial neural networks. What about the linearity of the device with a different doping concentration of the Si channel?

Response: We appreciate the reviewer for suggesting this insightful comment. It is valuable and indispensable to tune the device operation speed and power consumption for specific applications such as edge computing processors and high-performance processors.

Two types of GIFET were fabricated under different doping conditions. The doping concentration was controlled through the ion implantation process, which controls the length of the depletion layer and the overall resistance of the channel. The LTP-LTD characteristics of lightly and highly doped samples are shown in Fig. 3c and Supplementary Fig. 14a. The linearity of GIFET was similarly maintained even under different doping conditions (see Table R7). The doping concentration changes the drain current level, but linearity does not change significantly as shown in the table. This is because the linearity is determined by the thermionic emission through the gate stack, not by the doping concentration of the channel. We plotted the average of 10 repeated measurements with color.

	Lightly doping concentration	Highly doping concentration
Ion Implantation Condition (Doping Condition)	Dose : $2 \times 10^{13} \text{ cm}^{-2}$ Energy : 7.5 KeV Atom : Phosphorus	Dose : $5 \times 10^{14} \text{ cm}^{-2}$ Energy : 60 KeV Atom : Phosphorus
Silicon Channel Thickness	20 nm	145 nm
Linearity ($\alpha_{pot}/\alpha_{dep}$) (ideal = 1)	1.53 / 0.47	1.20 / -0.17
Device Measurement		
Table R7. Comparison of linearity with lightly and highly doping concentration.

- R1. Cheong, W. *et al.* A flash memory controller for 15µs ultra-low-latency ssd using high-speed 3d nand flash with 3µs read time. in *2018 IEEE International Solid-State Circuits Conference-(ISSCC)* 338–340 (IEEE, 2018).
- R2. Robertson, J. High dielectric constant oxides. *EPJ Appl. Phys.* **28**, 265–291 (2004).
- R3. Deb, S. K. Optical and photoelectric properties and colour centres in thin films of tungsten oxide. *Philos. Mag.* **27**, 801–822 (1973).
- R4. Kalema, V. N., Aljishi, S., Dawson, R. M. A., Slobodin, D. & Wagner, S. The dielectric constants of a-Si, Ge: H, F alloys. *Mater. Lett.* **4**, 320–322 (1986).
- R5. Li, Y. *et al.* Capacitor-based cross-point array for analog neural network with record symmetry and linearity. in *2018 IEEE Symposium on VLSI Technology* 25–26 (IEEE, 2018).
- R6. Yu, S. Neuro-inspired computing with emerging nonvolatile memories. *Proc. IEEE* **106**, 260–285 (2018).
- R7. Fuller, E. J. *et al.* Li- ion synaptic transistor for low power analog computing. *Adv. Mater.* **29**, 1604310 (2017).
- R8. Ambrogio, S. *et al.* Equivalent-accuracy accelerated neural-network training using analogue memory. *Nature* **558**, 60–67 (2018).
- R9. Zhang, W. *et al.* Neuro-inspired computing chips. *Nat. Electron.* **3**, 371–382 (2020).
- R10. Bhatt, V., Shrivastava, S., Chavan, T. & Ganguly, U. Software-Level Accuracy Using Stochastic Computing With Charge-Trap-Flash Based Weight Matrix. in *2020*

International Joint Conference on Neural Networks (IJCNN) 1–8 (IEEE, 2020).

- R11. Choi, J.-M., Park, E.-J., Woo, J.-J. & Kwon, K.-W. A highly linear neuromorphic synaptic device based on regulated charge trap/detrap. *IEEE Electron Device Lett.* **40**, 1848–1851 (2019).
- R12. Shrivastava, S., Chavan, T. & Ganguly, U. Ultra-low energy charge trap flash based synapse enabled by parasitic leakage mitigation. *arXiv Prepr. arXiv1902.09417* (2019).
- R13. Chen, P.-Y., Peng, X. & Yu, S. NeuroSim+: An integrated device-to-algorithm framework for benchmarking synaptic devices and array architectures. in *2017 IEEE International Electron Devices Meeting (IEDM)* 1–6 (IEEE, 2017).
- R14. Kwak, M. *et al.* Excellent Pattern Recognition Accuracy of Neural Networks Using Hybrid Synapses and Complementary Training. *IEEE Electron Device Lett.* **42**, 609–612 (2021).
- R15. Liu, X., Zheng, H., Li, Y. & Zhang, W. Factors on the separation of photogenerated charges and the charge dynamics in oxide/ZnFe₂O₄ composites. *J. Mater. Chem. c* **1**, 329–337 (2013).
- R16. Matsuura, H., Okuno, T., Okushi, H. & Tanaka, K. Electrical properties of n-amorphous/p- crystalline silicon heterojunctions. *J. Appl. Phys.* **55**, 1012–1019 (1984).
- R17. Ruder, S. An overview of gradient descent optimization algorithms. *arXiv Prepr. arXiv1609.04747* (2016).
- R18. Li, X. *et al.* Multi-terminal ionic-gated low-power silicon nanowire synaptic transistors with dendritic functions for neuromorphic systems. *Nanoscale* **12**, 16348–16358 (2020).
- R19. Zhu, J. *et al.* Ion gated synaptic transistors based on 2D van der Waals crystals with tunable diffusive dynamics. *Adv. Mater.* **30**, 1800195 (2018).
- R20. Kim, M. K. & Lee, J. S. Ferroelectric Analog Synaptic Transistors. *Nano Lett.* **19**, 2044–2050 (2019).
- R21. Yu, J. M. *et al.* All-Solid-State Ion Synaptic Transistor for Wafer-Scale Integration with Electrolyte of a Nanoscale Thickness. *Adv. Funct. Mater.* **2010971**, 1–10 (2021).
- R22. Go, J. *et al.* W/WO_{3-x} based three-terminal synapse device with linear conductance change and high on/off ratio for neuromorphic application. *Appl. Phys. Express* **12**, 26503 (2019).
- R23. Yang, C. Sen *et al.* All-Solid-State Synaptic Transistor with Ultralow Conductance for Neuromorphic Computing. *Adv. Funct. Mater.* **28**, 1–10 (2018).
- R24. Nikam, R. D., Kwak, M., Lee, J., Rajput, K. G. & Hwang, H. Controlled ionic tunneling in lithium nanoionic synaptic transistor through atomically thin graphene layer for neuromorphic computing. *Adv. Electron. Mater.* **6**, 1901100 (2020).
- R25. Yu, R. *et al.* Electret-based organic synaptic transistor for neuromorphic computing. *ACS Appl. Mater. Interfaces* **12**, 15446–15455 (2020).
- R26. Wang, L. *et al.* Exploring ferroelectric switching in α - In₂Se₃ for neuromorphic computing. *Adv. Funct. Mater.* **30**, 2004609 (2020).

- R27. Chung, W., Si, M. & Peide, D. Y. First demonstration of Ge ferroelectric nanowire FET as synaptic device for online learning in neural network with high number of conductance state and G_{\max}/G_{\min} . in 2018 IEEE International Electron Devices Meeting (IEDM) 12–15 (IEEE, 2018).
- R28. Chou, Y.-C. et al. Neuro-inspired-in-memory computing using charge-trapping memtransistor on germanium as synaptic device. *IEEE Trans. Electron Devices* **67**, 3605–3609 (2020).
- R29. Lee, K., Lee, J., Nikam, R. D., Heo, S. & Hwang, H. Sodium-based nano-ionic synaptic transistor with improved retention characteristics. *Nanotechnology* **31**, 455204 (2020).
- R30. Zhang, P. et al. Nanochannel-based transport in an interfacial memristor can emulate the analog weight modulation of synapses. *Nano Lett.* **19**, 4279–4286 (2019).
- R31. Fuller, E. J. et al. Li-Ion Synaptic Transistor for Low Power Analog Computing. *Adv. Mater.* **29**, 1–8 (2017).
- R32. Yeon, H. et al. Alloying conducting channels for reliable neuromorphic computing. *Nat. Nanotechnol.* **15**, 574–579 (2020).
- R33. Choi, S. et al. SiGe epitaxial memory for neuromorphic computing with reproducible high performance based on engineered dislocations. *Nat. Mater.* **17**, 335–340 (2018).
- R34. Wu, Q. et al. Improvement of durability and switching speed by incorporating nanocrystals in the HfO_x based resistive random access memory devices. *Appl. Phys. Lett.* **113**, 23105 (2018).
- R35. Jeon, Y.-R. et al. Suppressed stochastic switching behavior and improved synaptic functions in an atomic switch embedded with a 2D NbSe_2 material. *ACS Appl. Mater. Interfaces* **13**, 10161–10170 (2021).
- R36. Choi, S. H., Park, S.-O., Seo, S. & Choi, S. Reliable multilevel memristive neuromorphic devices based on amorphous matrix via quasi-1D filament confinement and buffer layer. *Sci. Adv.* **8**, eabj7866 (2022).

Reviewers' Comments:

Reviewer #1:

Remarks to the Author:

The authors have addressed all the issues and I recommend this paper to be accepted by Nature Communications now.

Reviewer #2:

Remarks to the Author:

Summary: The authors have shown the thermionic emission-based Gate Injection based Field-effect transistor. The authors have provided a lucid demonstration of the presence of thermionic emission rather than FN tunneling. The thermionic emission offers better linearity in comparison with tunneling-based three-terminal synaptic devices. However, the authors have not used appropriate methods to support their findings specifically for retention and variability. The literature survey for endurance of existing memories is incomplete. There are few major and minor comments to the author to improve the effectiveness of the manuscript.

1. The authors mentioned in Introduction that it is essential to maintain the retention at multiple states while training and inference are completed. We understand that the training is a dynamic phenomenon which completes in a time determined by the size of the training dataset but inference or recognition is a repeat process needed to be performed not just after training but much later as well (say after a year). The system has been trained to continue recognition/inference as and when new unseen input arrives. So it is not clear what "maintain retention while inference is completed means". Long-term inference cannot be performed in an analog drifting and low retention memory which is suitable only for training. The learnt weights need to be probably transferred to a non-volatile memory for usage in inference at any later stage.
2. The authors should check more literature on high endurance charge trap flash to modify their expectation of low CTF endurance, for example: Park, G. H., & Cho, W. J. (2010). Reliability of modified tunneling barriers for high performance nonvolatile charge trap flash memory application. *Applied Physics Letters*, 96(4), 043503., this work shows around 105 cycles endurance (same as that shown in Fig. 3(g) for the proposed device). And this number may greatly improve if the CTF device is used in an analog manner rather than digital switching. Please add appropriate reference to the line in Introduction: "However, owing to the endurance limit caused by the operation mechanism, research has been conducted on inference rather than online training." We do not believe CTF has as low an endurance as 103 as reported by the authors in their response.
3. In the Introduction: "These performances lead to the high accuracy of approximately 93.02% with MNIST handwritten recognition dataset, which is comparable to software baseline of 97%." These numbers are not representative of software baseline. MNIST has been solved in software with more than 99.9% accuracy (MNIST Benchmark (Image Classification) | Papers With Code). The reason the IEDM reference only shows 97% accuracy is that it used a binary neural network (1-bit weights) for its inference task. Also, a 93% accuracy is not comparable to 97% accuracy. We must think in terms of error. A 93% accurate (7% recognition error) network is more than twice as erroneous in classification as a 97% accurate (3% error).
4. Fig. 2d (1-8) are not relevant for demonstrating linearity. Any non-linear curve appears linear when zoomed in on a smaller region. Linearity has to be calculated over the entire dynamic range. Also, the linearity shown in Fig. 3(c) seems to be much worse compared to that shown in Fig. 2(c) even though the Fig. 3(c) is drawn for a smaller dynamic range of 5x current range. What is the true statistical non-linearity in these devices and at what dynamic range of conductance? What non-linearity and dynamic range has been used in the MNIST network training – do these numbers occur simultaneously (meaning good linearity at high dynamic range)? For example, Fig. 5(a) shows only a 2x dynamic range now but very good linearity.
5. What is the duration of the hold pulse in Fig. 2 (e)? Is it same as the write/erase pulse widths?
6. Why is the cycle to cycle and device to device variability reported using gate voltage in figures 3(a) and (b)? Since the output of the memory cell is drain current or conductance in all figures before, the variability in this output current should be reported at a particular read bias. The drain current will have especially high variability in subthreshold states of the MOSFET due to high voltage to current sensitivity. Reporting gate voltage variability is irrelevant to measured current output and the mode in which these devices are proposed to be used (current read). Hence the

variability numbers and benchmarking numbers need to be recalculated and their effect needs to be captured in the network training.

7. Why is the retention shown with a 8 V amplitude and 5 ms width pulse in Fig. 3(h) when this is not the operating voltage or pulse widths for any of the write-erase graphs? Also, the retention needs to be shown for different analog states. This state may have a good retention but what about another state. Atleast, choose an LRS and an HRS and show the retention of both states with time. The same is true for temperature retention measurements. Retention of different analog states needs to be added at every temperature and not one state with possibly good retention. Also, why do the drain current drift upwards with time for 393 K in Fig. 4(c)? There is a typo in the x-axis of Fig. 4(d) (323 K is repeated).

8. The endurance plot in Fig. 3(g) needs to be plotted with log x-axis to understand how the HRS/LRS change at different pulse numbers (low vs high), this is a standard methodology for showing logarithmic cycle numbers in endurance plots.

Reviewer #3:

Remarks to the Author:

The authors have largely addressed the comments. I would like to recommend the publication of this revised manuscript.

Response Letter to Reviewers' Comments

We sincerely appreciate the reviewers for investing their valuable time and effort in reviewing the manuscript and providing insightful comments and suggestions to further improve the quality of our work. Considering the reviewers' evaluations, we have made a point-by-point response to reviewer #2's comments and revised the manuscript and supplementary materials to improve the clarity of our work. We believe we have addressed reviewer #2's comments, and the paper is now more rigorous in content and clearer in presentation. Based on some responses below, we have revised or appended ten figures and one table in the revised manuscript, and two figures in the Supplementary Information to address the comments. Our point-by-point responses to the reviewers' comments are as follows:

Reviewer #1 (Remarks to the Author):

The authors have addressed all the issues and I recommend this paper to be accepted by Nature Communications now.

Response: We highly appreciate the reviewer for the valuable time for reviewing our manuscript and constructive comments to help significantly improve the quality of our work.

Reviewer #2 (Remarks to the Author):

Summary: The authors have shown the thermionic emission-based Gate Injection based Field-effect transistor. The authors have provided a lucid demonstration of the presence of thermionic emission rather than FN tunneling. The thermionic emission offers better linearity in comparison with tunneling-based three-terminal synaptic devices. However, the authors have not used appropriate methods to support their findings specifically for retention and variability. The literature survey for endurance of existing memories is incomplete. There are few major and minor comments to the author to improve the effectiveness of the manuscript.

Response: We sincerely thank the reviewer for providing constructive comments on our revised manuscript. Based on the reviewer's comments and suggestions, we also measured our GIFET to supplement previous data, especially in terms of retention and variability. We revised the manuscript by showing robust data in terms of reliability and uniformity, and we performed the MNIST image classification simulation again with the new dataset. In addition, we restated the roles of DRAM and charge trap flash in terms of neuromorphic computing from other literature and clarified the advantages of GIFET. Based on the reviewer's valuable comments, the readability of the manuscript has been improved to prevent misunderstandings. Our detailed

responses to the reviewer's comments are provided below.

Comment #1

The authors mentioned in Introduction that it is essential to maintain the retention at multiple states while training and inference are completed. We understand that the training is a dynamic phenomenon which completes in a time determined by the size of the training dataset but inference or recognition is a repeat process needed to be performed not just after training but much later as well (say after a year). The system has been trained to continue recognition/inference as and when new unseen input arrives. So it is not clear what “maintain retention while inference is completed means”. Long-term inference cannot be performed in an analog drifting and low retention memory which is suitable only for training. The learnt weights need to be probably transferred to a non-volatile memory for usage in inference at any later stage.

Response: We would like to thank the reviewer for this valuable comment. As the reviewer mentioned, the meaning of “maintain retention while inference is completed” in the Introduction is unclear, and it seems inappropriate to use this sentence in the context of the limitations of DRAM compared to GIFET. Although a memory device with excellent endurance characteristics but poor retention like DRAM is not suitable for an inference that needs to store weight values without data loss for a long period of time, it is much more suitable for training, which is a repetitive dynamic phenomenon. This is because the weights are frequently updated during online training, high robustness in terms of endurance is required and, conversely, data retention requirements can be partially alleviated^{R1-R4}. The trained weight values simply need to be moved and stored in memory devices with decent nonvolatile properties.

The difference between DRAM and the proposed device is the retention time. Capacitor-based synaptic devices with a DRAM-like structure usually store data for several to tens of seconds without a loss in the ref. R5, but GIFET can maintain multi-level conductance states stably for more than 1,000 seconds, as shown in Fig. R1. This means that the data transfer will occur less with the proposed GIFET than with the DRAM. Furthermore, in the case of the DRAM-like capacitor-based device, several additional transistors are required to implement analog conductance states^{R3,R5,R6}. Whereas, multiple conductance levels with a high degree of integration density can be achieved in a single cell with the GIFET.

Fig. R1 Retention characteristics of GIFET at multi-level conductance states. All the measured multi-level conductance states were maintained for 1,000 s without noticeable conductance loss (pulse condition: [-3.3 V, 500 μ s to the gate for reset], [5 V, 500 μ s to the gate for set], [1 V, 50 μ s to the drain for read]).

In conclusion, the proposed device cannot avoid data transfer for weight storage to the long-term memory, as the reviewer pointed out, but it can frequently reduce data transfer and improve energy efficiency. We are now seeking to improve the retention characteristics of the device to be used for inference in future work. The sentence “maintain retention while inference is completed” has been removed and we have added a sentence to emphasize the advantages of low power consumption due to fewer data transfers during training and the potential to integrate devices with high density.

We revised the sentences to reflect the reviewer’s comment and to avoid confusion in the manuscript on page 3, line 46:

“In the case of conventional memories, the well-established DRAM secures fast write speed and linear conductance update¹⁰. Capacitor-based synaptic devices with a DRAM-like structure also have main advantages in online training for repeated updates because of their high endurance¹⁸. However, because of poor retention characteristics, the weight values must be transferred to nonvolatile memories very frequently during the training process, resulting in high power consumption. Moreover, these devices are difficult to create and retain analog conductance states with a single device and require a capacitor (storing charges for weight values) and several additional transistors to implement analog states^{10,19,20}. This means that it has a drawback in terms of device integration density compared to a single synaptic device.”

Comment #2

The authors should check more literature on high endurance charge trap flash to modify their expectation of low CTF endurance, for example: Park, G. H., & Cho, W. J. (2010).

Reliability of modified tunneling barriers for high performance nonvolatile charge trap flash memory application. Applied Physics Letters, 96(4), 043503., this work shows around 10^5 cycles endurance (same as that shown in Fig. 3(g) for the proposed device). And this number may greatly improve if the CTF device is used in an analog manner rather than digital switching. Please add appropriate reference to the line in Introduction: “However, owing to the endurance limit caused by the operation mechanism, research has been conducted on inference rather than online training.” We do not believe CTF has as low an endurance as 10^3 as reported by the authors in their response.

Response: We appreciate the reviewer for providing this thoughtful comment and suggesting helpful references. We previously stated that charge trap flash memory (CTF) was mainly studied for inference rather than training because of its endurance characteristic limitations due to the operating mechanism. However, thanks to the reviewer’s comments and references, we realized that the previous statement was inappropriate and we revised the statement as shown below.

In many studies on neuromorphic computing devices, endurance characteristics are emphasized with repetitive data writing and erasing operations for online training. In contrast, retention characteristics of nonvolatile memory are crucial for inference because the synaptic weights should be secured for long periods after offline training^{R3,R7,R8}. In particular, endurance should be at least 10^6 or more for stable online training^{R9}. Previously, we used the endurance of 10^4 cycles from commercialized conventional NAND flash memory. Based on this number, it was considered that flash memory is suitable only for inference rather than for online training. However, in other studies, including the papers provided by reviewer #2, the endurance characteristics of CTF were significantly improved through structural engineering and material modification^{R10-R12}. Furthermore, there was an attempt at online training by using CTF directly^{R13}. However, compared to other commercial memories or emerging memories that are currently being studied, CTF generally requires high update voltages and has a slow operation speed^{R4,R14,R15}. These factors consequently cause high energy consumption, which could create drawbacks for online training.

To eliminate the misunderstanding of the phrase that CTF has been studied only for inference in the neuromorphic computing field, and to clarify the advantages and drawbacks of CTF, the following paragraph in the manuscript was revised on page 3, line 55:

“On the other hand, nonvolatile memories such as CTF can distinguish between states of multi-level cells depending on how many charges are trapped in the charge trap layer, and have long retention¹². Therefore, research using CTF devices is being actively conducted for applications in neuromorphic computing, as well as for the main memory for data storage. However, while data can be stored for long periods without data loss, CTF normally has a large operation voltage and slow speed, requiring more energy, especially for data erasing²¹⁻²³. In the case of online training, using devices with low update energy is advantageous because the training demands repeated writing and erasing operations more than millions of times.”

In addition, the extra endurance characteristics were measured to confirm that sufficient repetitive write/erase operations are possible for online training of GIFET. The write/erase pulse condition is 5 V, 200 μ s/-5 V, 200 μ s, and 2×10^8 pulses in total (2×10^5 cycles for LRS/HRS) were verified as shown in Fig. R2. We replaced Fig. 3g with Fig. R2 to show better endurance characteristics.

Fig. R2 Endurance characteristics of GIFET. Write/erase pulses were applied to the gate 500/500 times for one cycle, and the measurement was conducted for 2×10^8 update pulses (2×10^5 cycles). The condition of the write/erase pulse is 5 V, 200 μ s/-5 V, 200 μ s, and read operation (1 V, 50 μ s) followed by every write and erase pulse.

We have revised some sentences that state the endurance characteristics in the manuscript:

Page 12, line 223: “The endurance of GIFET over 2×10^5 switching cycle (2×10^8 pulses). Each switching cycle is composed of 500 write pulses with 5 V, 200 μ s and 500 erase pulses with -5 V, 200 μ s.”

Page 14, line 261: “The endurance and retention of the device are also crucial for long-term and reliable neuromorphic computing applications⁴⁹. To investigate the endurance of GIFET, we applied 500 consecutive potentiation pulses with an amplitude of 5 V and a width of 200 μ s, followed by 500 consecutive depression pulses with an amplitude of -5 V and a width of 200 μ s per switching cycle. We then read the change in state by drain voltage (1 V, 50 μ s) at each switching cycle. As presented in Fig. 3g, the device achieves robust endurance ($\geq 2 \times 10^8$ pulses).”

Comment #3

In the Introduction: “These performances lead to the high accuracy of approximately 93.02% with MNIST handwritten recognition dataset, which is comparable to software baseline of 97%.” These numbers are not representative of software baseline. MNIST has

been solved in software with more than 99.9% accuracy (MNIST Benchmark (Image Classification) | Papers With Code). The reason the IEDM reference only shows 97% accuracy is that it used a binary neural network (1-bit weights) for its inference task. Also, a 93% accuracy is not comparable to 97% accuracy. We must think in terms of error. A 93% accurate (7% recognition error) network is more than twice as erroneous in classification as a 97% accurate (3% error).

Response: We are grateful to the reviewer for providing this insightful comment. As the reviewer mentioned, we agree that the software baseline of 97% is not representative of the software-based MNIST classification, and the accuracy of ‘97%’ and the approximately ‘93%’ accuracy of the simulation result of GIFET cannot be comparable. Also, referring to the presented MNIST Benchmark (Image Classification) – Paper With Code, the simulation models used for MNIST classification are ranked in the order of the lowest percentage error, and the model that achieved the highest accuracy (lowest error) shows an accuracy of 99.91% (0.09% error). In the case of the high-accuracy models shown in this benchmark, the top three models (heterogeneous ensemble with simple CNN^{R16} [99.91%], branching/merging CNN + homogeneous vector capsules^{R17} [99.87%], EnsNet^{R18} [99.84%]) were mainly performed based on a convolutional neural network (CNN) structure. The accuracy of MNIST classification is significantly affected by what factors, such as learning algorithms and parameters, are used. We calculated the classification accuracy reflecting the characteristics of GIFET in this work by using NeuroSim+ simulation^{R19}. This neural network consists of three layers and performs supervised learning with back propagation. The input layer has 400 nodes for 20×20 pixels of the binary (black and white) MNIST images, and the hidden neuron has 100 nodes, while the output neuron has ten nodes for the classification results, representing 0–9 digits. Each weight of the synaptic device was updated using the stochastic gradient descent method. That is, the multi-level perceptron (MLP) model provided by NeuroSim+ is a very simple artificial neural network (ANN) structure. Moreover, the simulation with GIFET does not intend to prove the high-performance image classification ability of the neural network, but to prove the effect of the device characteristics (linearity, variations, etc.) on classification accuracy.

For a fair comparison, it will be necessary to compare the simulation results of GIFET with those of other synaptic devices using the neural network with the same or almost identical structure. The reason for referring to the accuracy of 97% was that the structure of the neural network performed in the ref. R20 (400 nodes for 20×20 pixels of binary MNIST images, 200 nodes for the hidden layer, and ten nodes for the output neuron) was similar to our work. However, it is still unreasonable to compare the results of the GIFET simulation with those of the reference, since MLP structures are not perfectly identical. Moreover, it is not appropriate to compare the two results because there is a huge gap in terms of classification error between the accuracy of 93% and that of 97%. Instead, considering several parameters, as shown in Table R1 (also suggested in Supplementary Table 1), the GIFET performance can be compared with other synaptic transistors in the same MLP structure.

We have revised the manuscript as follows to reflect the reviewer’s comment and to delete

inappropriate comparisons:

Page 4, line 86: “In this paper, we propose a three-terminal Gate Injection-based Field-Effect Transistor (GIFET), which utilizes the CMOS compatible material and fabrication process. Through different operation mechanisms from conventional flash memory, we derive superior synapse device characteristics, such as high linearity and symmetry, high temporal and spatial uniformity (< 1.64%, 9.76%), and low power consumption (50 fJ/SOP). These performances lead to a high accuracy of approximately 93.17% with the MNIST handwritten recognition dataset.”

	[R21]	[R22]	[R23]	[R24]	[R25]	This work
Multi Layer Perceptron	400 input 100 hidden 10 output	400 input 100 hidden 10 output	400 input 100 hidden 10 output	400 input 100 hidden 10 output	400 input 100 hidden 10 output	400 input 100 hidden 10 output
Conductance states (Potentiation / Depression)	64 / 64	50 / 50	100 / 100	320 / 256	35 / 35	1,000 / 1,000
Nonlinearity (Ideal = 0) (LTP / LTD)	-1.54 / -1.76 (Not use identical pulses)	0.07 / -2.42	N/A (Not use identical pulses)	1.22 / -1.75	0.06 / -0.89 (Not use identical pulses)	0.96 / -0.89
G_{\max}/G_{\min}	> 10	> 100	5	> 100	> 10	> 10
Temporal Variation	YES	No	YES	YES	No	YES
Spatial Variation	YES	No	No	No	No	YES
Accuracy	91.1%	85.88%	90.6%	88%	87%	93.17%

Table R1. Comparison of the accuracy and parameters reflected in the MNIST simulation. This table is identical to the table in Supplementary Table 1.

Comment #4

Fig. 2d (1-8) are not relevant for demonstrating linearity. Any non-linear curve appears linear when zoomed in on a smaller region. Linearity has to be calculated over the entire dynamic range. Also, the linearity shown in Fig. 3(c) seems to be much worse compared to that shown in Fig. 2(c) even though the Fig. 3(c) is drawn for a smaller dynamic range of 5x current range. What is the true statistical non-linearity in these devices and at what dynamic range of conductance? What non-linearity and dynamic range has been used in the MNIST network training – do these numbers occur simultaneously (meaning good linearity at high dynamic range)? For example, Fig. 5(a) shows only a 2x dynamic range now but very good linearity.

Response: We would like to thank the reviewer for suggesting constructive comments. As the reviewer mentioned, in all LTP-LTD plots, even nonlinear shapes, zooming in on some areas can make them appear linear. Figure 2d, however, was intended to emphasize the linearity in

the entire conductance levels by showing the changes in conductance with the same number of pulses rather than showing a linear slope by enlarging some specific regions. For the training process, we need to update the conductance at each training step based on Δw . If the device shows nonlinearity in the entire region, which means the number of pulses = $f(\Delta w, \text{current } w)$, we need to apply the read process to check the current status of the device. However, the GIFET shows linearity in the entire region, which means the number of pulses = $f(\Delta w)$, we do not need to apply the read process after each training process. To prevent misunderstanding, we added conductance changes at each of the 100 pulses, as shown in Fig. R3.

Fig. R3 LTP-LTD characteristics of GIFET under 1,000 write-1,000 erase consecutive pulse schemes. Each write and erase pulse is composed of 5 V, 500 μs and -3.3 V, 500 μs , respectively. Read operation (1 V, 50 μs) was conducted between every write/erase operation. The left figure shows the entire linear conductance update and the right eight small figures show the conductance changes during every 100 pulses, showing almost the same conductance changes at each region.

In addition, as the reviewer mentioned, there are differences in write/erase pulse condition, linearity, and dynamic range of conductance update for each LTP-LTD plot in the manuscript. Specifically, Fig. R4a and b (identical to Fig. 2c and Fig. 3c) show the LTP-LTD characteristics of the GIFET. Since the pulse conditions are different in each case, the linearity and dynamic range are also different. To avoid confusion, LTP-LTD characteristics were measured again with the same pulse conditions as in Fig. R4a.

For the simulation, we extracted the parameters described below.

Basically, Fig. R4a is the representative LTP-LTD characteristics of GIFET and is the basis for ‘true statistical nonlinearity and dynamic range of conductance’. Nonlinearity was 0.96 and -0.89 in LTP and LTD, respectively, and these values were calculated by the method described in Supplementary Note 2. The dynamic range, which was approximately 10, was also set based on the experimental data from Fig. R4a, G_{max} and G_{min} are 19.81 nS and 206.90 nS, respectively. Operation pulse schemes were 5 V, 500 μs for write, and -3.3 V, 500 μs for erase.

Cycle-to-cycle and device-to-device variations were newly measured under the same pulse

conditions as in Fig. R4a. Fig. R4c shows the new LTP-LTD characteristic of GIFET for 100 cycles in the same single device, and the cycle-to-cycle variation (σ/μ) is 1.64%. The device-to-device variation remeasured for 15 different devices with the same gate width and length is shown in Fig. R4d. The device-to-device variation is 9.76%, and since the standard deviation of nonlinearity of each LTP-LTD plot should be applied to the NeuroSim+ simulation, the average value of ~ 0.40 was used. This standard deviation of nonlinearity is explained in comment 6 of reviewer #2.

Based on the newly measured variation data, the MNIST network simulation was performed again. In conclusion, the ‘true statistical nonlinearity and dynamic range of conductance’ of GIFET are 0.96/-0.89, and ~ 10 , respectively. Furthermore, these factors can be obtained simultaneously from the same LTP-LTD plot.

Fig. R4 LTP-LTD characteristics of GIFET and its cycle-to-cycle and device-to-device variations. **a** LTP-LTD characteristics of GIFET under 1,000 write (5 V, 500 μ s)-1,000 erase (-3.3 V, 500 μ s) consecutive pulse scheme (1 V, 50 μ s read operation and 50 μ s between each pulse were used). **b** Ten repeated LTP-LTD characteristics on a single GIFET device with 1,000 potentiation-1,000 depression (1.4 V/-2.5 V, 500 μ s). Fig. R4a and b are identical to Fig. 2c and Fig. 3c, respectively. **c** 1.64% of cycle-to-cycle variation (σ/μ) measured from 100 repeated cycles in a single device. **d** 9.76 % of device-to-device variation (σ/μ) measured from 15 different devices.

Lastly, Fig. R5 (the same as Fig. 5a in the manuscript) shows a different pulse condition from the other LTP-LTD plots. The pulse scheme and plot in Fig. R5 are intended to show the difference in the read process with conventional flash memory, not the verification for linearity and dynamic range of GIFET. In flash memory, after the operating conductance level of the transistor is determined according to charge trapping, a voltage should be applied to the gate during the read operation and a read voltage is applied to the drain to read the conductance state. However, since GIFET is a junctionless transistor structure, the conducting channel is maintained even when 0 bias is applied to the gate. The conductance state is caused by a change in the size of the depletion region according to the negative charge in the charge store layer (CSL). Specifically, as in the range of wait pulses (number 500~600 and 1,700~1,800) in Fig. R5, even when the gate bias is 0 V, the conductance state is secured stably.

Fig. R5 LTP-LTD characteristics of GIFET and its operation pulse scheme. The graph shows the channel conductance update with 1 V, 30 μ s read pulses (write/erase with 1.8 V/-2.5 V, 300 μ s). The conductance state was secured stably during the holding process ($V_d=1$ V, $V_g=0$ V) where the wait pulse number is 500~600 and 1,700~1,800. This figure is identical to Fig. 5a.

To reflect the newly measured data, we revised Fig. 2 and 3 in the manuscript:

“

Figure 2. The linearity of GIFET and its relationship with the conduction mechanism. a Schematic of the GIFET gate stack without gate oxide and band diagram of CSL and blocking layer during write operation. The barrier exists between the CSL and the blocking layer, and the charge in the CSL is extracted over the barrier. **b** The Arrhenius plot measured from WO_x/a-Si:H stack without the SiO₂ layer between the channel and the stack from 273 K to 423 K. **c** The LTP-LTD characteristic of GIFET under 1,000 write (5 V, 500 μ s)-1,000 erase (-3.3 V, 500 μ s) consecutive pulse scheme (1 V, 50 μ s read voltage and 50 μ s between each pulse was used). **d** Linear write (red sphere)/erase (blue sphere) update of the LTD-LTD during every 100 pulses in specific ranges matched the numbers shown in Fig. 2c. Almost similar drain current increased/decreased within the same number of pulses. **e** The LTP-LTD characteristics of the GIFET under arbitrary pulse trains (haphazard write/erase pulse) consist of 500 write (1.4 V, 500 μ s)/erase (-2.5 V, 500 μ s)/hold (0 V, 500 μ s) gate pulses with a read pulse (1 V, 500 μ s) on

the drain 200 μ s after each gate pulse (red: write operation, blue: erase operation, white: hold operation). **f** Stable linear update values of Fig. 2e (E-W-W-E) ($\Delta I_D = 5.71$ nA, 5.78 nA).

Fig. 3 Measured GIFET data for high performance in a crossbar-array structure. a, b Selected gate voltages based on reaching a constant current (3 nA) in I_D - V_G characteristics. Gate voltage was double swept from -4 V to 7 V, and 50 cycles in a single device (**a**) and 15 different devices with the same gate width and length (**b**) were measured. Insets are based on data from Supplementary Fig. 12. **c** Repeated LTP-LTD characteristics on a single GIFET device with 1,000 potentiation–1,000 depression (5 V/-3.3 V, 500 μ s). **d** LTP-LTD characteristics for 15 different devices with 1,000 potentiation–1,000 depression (5 V/-3.3 V, 500 μ s). The cycle-to-cycle and device-to-device variations of GIFET are 1.64% and 9.76%, respectively. **e** I_G - V_G characteristic of the GIFET. **f** LRS current level of GIFET in I_D - V_G characteristic (inset) according to channel doping concentration. **g** The endurance of GIFET over 2×10^5 switching cycles (2×10^8 pulses). Each switching cycle is composed of 500 write pulses with 5 V, 200 μ s and 500 erase pulses with -5 V, 200 μ s. **h** Retention characteristic of GIFET. Erase pulses (-3.3 V, 500 μ s) were applied to the gate to reach a certain conductance level, and read pulses (1 V, 50 μ s) were applied to the drain every 1 s after the erase pulse train.

A 5.45% conductance value change was observed after 1,000 s.”

We have also revised the manuscript:

Page 10, line 188: “As shown in Fig. 2d, the device shows stable linear conductance updates in the entire conductance state, which means it has similar conductance changes with the same number of pulses.”

Comment #5

What is the duration of the hold pulse in Fig. 2 (e)? Is it same as the write/erase pulse widths?

Response: We appreciate the reviewer for providing this insightful comment. The write/erase/hold pulse conditions shown in Fig. 2e were all performed under the same conditions (duration: 500 μ s).

We have revised the sentences related to Fig. 2e in the manuscript:

Page 9, line 146: “The LTP-LTD characteristics of the GIFET under arbitrary pulse trains (haphazard write/erase pulse) consists of 500 write (1.4 V, 500 μ s) / erase (-2.5 V, 500 μ s) / hold (0 V, 500 μ s) gate pulses with a read pulse (1 V, 500 μ s) on the drain 200 μ s after each gate pulse (red: write operation, blue: erase operation, white: hold operation).”

Page 11, line 198: “Figure 2e presents the linear conductance update with arbitrary pulse trains. Pulses consisting of 500 write pulses (1.4 V, 500 μ s), 500 hold pulses (0 V, 500 μ s), and 500 erase pulses (-2.5 V, 500 μ s) were applied to the gate. For the read process, a read pulse (1 V, 200 μ s) was applied to the drain terminal.”

Comment #6

Why is the cycle to cycle and device to device variability reported using gate voltage in figures 3(a) and (b)? Since the output of the memory cell is drain current or conductance in all figures before, the variability in this output current should be reported at a particular read bias. The drain current will have especially high variability in subthreshold states of the MOSFET due to high voltage to current sensitivity. Reporting gate voltage variability is irrelevant to measured current output and the mode in which these devices are proposed to be used (current read). Hence the variability numbers and benchmarking numbers need to be recalculated and their effect needs to be captured in the network training.

Response: We appreciate the reviewer’s suggestion of this constructive comment. Figures R6a and b (the same as Fig. 3a and b, respectively) are data extracted from Supplementary Fig. 12, I_D - V_G characteristics of GIFET. These two figures show gate voltages reaching the specific constant drain current (3 nA) in the transfer curve of GIFET with hysteresis. The low resistive state (LRS) and high resistive state (HRS) shown in the figures refer to two cases: the former means that electrons are removed from the CSL and thus the channel resistance is low due to the small depletion region, while the latter means CSL is charged with electrons and thus the channel resistance is high owing to the large depletion region. This means that the latter requires a larger gate voltage than the former to turn on the transistor. We intended to calculate the cycle-to-cycle and device-to-device variations of GIFET by utilizing the method in which a state changes based on a specific threshold. For example, in the case of 2-terminal devices such as ReRAM, the set or reset voltage converting HRS to LRS or LRS to HRS can be determined by the voltage when the current measured in a voltage sweep reaches a specific threshold current. For a better understanding of the figures, we modified them as Fig. R7a and b.

Fig. R6 Extracted gate voltage distribution from I_D - V_G characteristics of GIFET. The gate voltages are selected to reach the specific constant current (3 nA) in the transfer curve of GIFET from Supplementary Fig. 12. **a** Gate voltage variation for the repeated I–V sweep of 50 cycles in a single device. **b** Gate voltage variation for 15 different devices with the same gate length and width. The two figures are identical to Fig. 3a and b in the manuscript before the revision.

Fig. R7 Modified figures for a better understanding of Fig. R6 (Fig. 3a and b in the manuscript). Insets are **a** repeated I–V sweep of 50 cycles in a single device, and **b** 15 different devices with the same device geometry.

However, as the reviewer mentioned, the electrical characteristics of GIFET in most of the figures shown in the manuscript were verified by applying a specific read voltage and measuring the drain current. For this reason, the cycle-to-cycle and device-to-device variations were recalculated by measuring the LTP-LTD characteristics of repetitive pulse cycles and multiple devices. As shown in Fig. R8, the new LTP-LTD characteristics were measured in the same pulse condition as in Fig. 2c (write: 5 V, 500 μs , erase: -3.3 V, 500 μs , read: 1 V, 50 μs). Fig. R8a shows the measurement results of repeating 100 cycles of 1,000 write-1,000 erase operations on a single device, and the average cycle-to-cycle variation is 1.64%, which has high temporal uniformity. In addition, the device-to-device variation is 9.76% for 15 devices with the same device geometry, as shown in Fig. R8b.

Unfortunately, because GIFET was fabricated using university-level fabrication processes, the quality of deposited films may be inferior to that of CMOS-based foundry processes. In addition, because GIFET in this study is a junctionless structure, the drain current is dominantly determined by the thickness of the silicon channel, which causes spatial variation. Therefore, we expect that device-to-device variation can be improved by using a better quality thin film process and conventional channel inversion MOSFET operation.

Fig. R8 Cycle-to-cycle and device-to-device variations of GIFET in LTP-LTD characteristics. **a** Repeated LTP-LTD characteristic on a single GIFET device with 1,000 potentiation – 1,000 depression (5 V/-3.3 V, 500 μ s). **b** LTP-LTD characteristics for 15 different devices with 1,000 potentiation – 1,000 depression (5 V/-3.3 V, 500 μ s). The cycle-to-cycle and device-to-device variations of GIFET is 1.64% and 9.76%, respectively.

Based on the above results, we performed the MNIST classification again using NeuroSim+ by reflecting the new measured cycle-to-cycle and device-to-device variation results. To reflect the device-to-device variation as GIFET's characteristics in NeuroSim+ simulations, the nonlinearities of each LTP-LTD plot should be calculated. The device-to-device variation was defined as the standard deviation of the nonlinearity baseline in the NeuroSim+ simulation. The following equation was used to calculate the nonlinearity of the GIFET in the simulation:

$$G_{LTP} = B \left(1 - e^{\frac{P}{A}} \right) + G_{min}$$

$$G_{LTD} = B \left(1 - e^{\frac{P - P_{max}}{A}} \right) + G_{max}$$

$$B = (G_{max} - G_{min}) / \left(1 - e^{\frac{-P_{max}}{A}} \right)$$

where G_{LTP} and G_{LTD} are the conductance of LTP and LTD, respectively. G_{max} , G_{min} , and P_{max} were directly extracted from the experimental data. A is a parameter that controls the nonlinear behavior of the weight update. This value is close to 0, which represents ideal linearity.

Fig. R9a and b show the nonlinearities extracted using the data from the measured LTP-LTD plots of each of the 15 devices in Fig. R8b and the above equation (same with Supplementary Note 2). The standard deviation of nonlinearity (σ_{NL}) of GIFET was calculated as 0.29/0.51 for potentiation and depression, respectively. By reflecting the new measured data of cycle-to-cycle and device-to-device variations, we obtained an accuracy of approximately 93.17% with the real measured device characteristics during 300 epochs, as shown in Fig. R9c. The case of the ideal device (software baseline simulation with ideal circumstances) and ideal linearity (reflecting all the characteristics of GIFET except linearity and device variations) showed an

accuracy of 96.78% and 93.45%, respectively. This means that there is a 0.28% loss of accuracy due to the linearity and device variation of GIFET. It confirms that the linearity and variation factors of GIFET do not degrade the accuracy of MNIST classification.

Fig. R9 Effect of variations on the LTP/LTD. **a, b** Normalized LTP-LTD characteristics based on Fig. R8b. **a** Potentiation device variation under 1,000 potentiation pulses (5 V/500 μ s). **b** Depression device variation under 1,000 depression pulses (-3.3 V / 500 μ s). **c** Simulation accuracy of the MNIST image classification reflecting new measured results of cycle-to-cycle and device-to-device variations in the LTP-LTD. The simulation was conducted for five cases: ideal device, ideal linearity, real GIFET, lightly-deteriorated linearity, and highly-deteriorated linearity.

To reflect the reviewer’s comment, we have revised the figures and sentences regarding device variations in the manuscript:

Page 13, line 234: “First, to investigate the spatio-temporal uniformity of the GIFET, we assessed I_D - V_G characteristics of the GIFET by gate voltage sweeping with constant read voltage on the drain. Figures 3a and b present gate voltage at a specific drain current ($I_D = 3$ nA) from repeated cycles on a single device for cycle-to-cycle variation and from 15 different devices for device-to-device variation, respectively. In these figures, cycle-to-cycle variation was observed as 4.30% at HRS and 1.15% at LRS (σ/μ), while device-to-device variation was measured as 5.16% at HRS and 3.67% at LRS (σ/μ) (ref. Supplementary Fig. 12). In addition, the repeated LTP-LTD characteristics on a single device and several different devices were observed with 1,000 potentiation (5 V, 500 μ s)–1,000 depression (-3.3 V, 500 μ s) gate pulse trains, as shown in Fig. 3c and d. The LTP-LTD characteristics of the device in Fig. 3c presented a low variation of 1.64 % (σ/μ) for 100 repeated cycles. The device-to-device variation was experimentally measured on 15 devices and it showed 9.76% (see Fig. 3d). The standard deviation of nonlinearity based on the LTP-LTD characteristics of 15 devices is also calculated in Supplementary Fig. 13. The nonlinearity during potentiation/depression in Supplementary Fig. 13a and b was fitted using the method from Supplementary Note 2. Each spatial variation in the results of the above I_D - V_G and LTP-LTD characteristics was slightly higher than each temporal variation because of the Si channel thickness variation during fabrication (see Methods section). This structure shows uniform switching because it utilizes a large population

of electrons, minimizing the effect of fluctuation or stochastic behavior of individual charged particles, instead of individual ion movement.”

We revised Fig. 5c and the sentences about the simulation results:

Page 17, line 309:

“

Fig. 5 GIFET simulation with MNIST dataset.”

Page 17, line 313: “MNIST image classification simulation to compare the GIFET characteristics with ideal device (software baseline simulation with ideal circumstances such as large G_{max}/G_{min} ratio, ideal linearity, and no device variations), ideal linearity (reflecting all the characteristics of GIFET except linearity and device variations) and the devices with deteriorated linearity (nonlinearity of 3, 6 for lightly-, highly-deteriorated linearity, respectively).”

Page 19, line 344: “As observed, the ideal device shows an accuracy of 96.78%, and the GIFET-based artificial neural network obtained an accuracy of approximately 93.17% during 300 epochs, which is almost equivalent to that of an ideal linearity device, 93.45%. Furthermore, the device with lightly- and highly-deteriorated linearity exhibited degraded training results as maximum accuracy of 85.13% and 70.56%, respectively.”

We have also revised Supplementary Fig. 13 to reflect the new experimental data:

“

Supplementary Fig. 13. Device-to-device variation of LTP-LTD characteristics. a, b Normalized LTP-LTD characteristics of 15 different devices extracted from Fig. 3d. Nonlinearity was fitted using the method from Supplementary Note 2. The average standard deviations of the nonlinearity ($\sigma_{NL, pot}$, $\sigma_{NL, dep}$) were calculated for simulation of MNIST classification based on Neurosim+.”

Lastly, we changed the simulation accuracy of ‘93.02%’ in the manuscript to ‘93.17%’ to reflect the new simulation result.

Comment #7

Why is the retention shown with a 8 V amplitude and 5 ms width pulse in Fig. 3(h) when this is not the operating voltage or pulse widths for any of the write-erase graphs? Also, the retention needs to be shown for different analog states. This state may have a good retention but what about another state. Atleast, choose an LRS and an HRS and show the retention of both states with time. The same is true for temperature retention measurements. Retention of different analog states needs to be added at every temperature and not one state with possibly good retention. Also, why do the drain current drift upwards with time for 393 K in Fig. 4(c)? There is a typo in the x-axis of Fig. 4(d) (323 K is repeated).

Response: We appreciate the reviewer’s insightful comments. We agree with the reviewer’s suggestion and performed the measurement again. As the reviewer mentioned, we performed retention in the multi-level conductance levels with identical pulse conditions from other measurements. We also performed the temperature-dependent measurement again.

Accordingly, we remeasured the retention characteristics in various conductance states, and the measurement was performed in the same way as the write/erase conditions of the other measured LTP-LTD data. The retention characteristics of the four different conductance states

for 1,000 s are shown in Fig. R10b. Erase pulses (-3.3 V, 500 μ s) of 100, 400, 700, and 1,000 were applied to the gate, and the drain current was measured every 1 s by applying read pulses (1 V, 50 μ s) to the drain after the erase pulse trains. As shown in Fig. R10b, 13.6 nS (5.45%) of conductance was changed in the case of the largest change during 1,000 s, and the states were unchanged in the other cases.

Fig. R10 Previous retention data (removed from the revised manuscript) and newly measured retention characteristics of multi-level conductance states. **a** Previous retention characteristic of GIFET. After a 1,500 set pulse train with 8 V amplitude and 5 ms width was applied, the drain current was measured every 0.4 s while the gate was grounded. A 7.15% updated data loss was observed after 1,200 s. This figure is identical to Fig. 3h. **b** Multi-level conductance states of GIFET. Erase pulses (-3.3 V, 500 μ s) were applied to the gate to reach a certain conductance level, and read pulses (1 V, 50 μ s) were applied to the drain every 1 s subsequent to the erase pulse train. A 5.45% updated data loss was observed after 1,000 s.

To verify the reliability of multi-level conductance states at high temperatures, the retention characteristics according to temperature (from 298 K to 393 K) were measured again for the two states (see Fig. R11). As shown in Fig. R11b, erase pulses (-3.3 V, 500 μ s) of 100 (red sphere) and 1,000 (blue sphere) were applied in the initial state, respectively, and read operations (1 V, 50 μ s) were performed every 1 s after the erase pulse train. As a result, it was confirmed that only the conductance level (drain current level) increased as the temperature increased and that the conductance states remained unchanged for 200 s. In the previous retention data (see Fig. R11a), especially at 393 K, the drain current drifted upward without maintaining the data over time. We believe that this phenomenon was caused by thermionic emission because the charge storage mechanism of GIFET is based on the energy barrier between CSL and the blocking layer, and electrons at high temperatures can escape CSL more easily than electrons at room temperature. However, the data of the newly fabricated device maintained conductance states for a longer time, even at a high temperature of 393 K. That is, it is believed that the quality of the deposited thin films predominantly affects retention characteristics.

Fig. R11 Retention characteristics measurement at various temperatures from 298 K to 393 K **a** Previous retention characteristics at various temperatures. Depression pulses (-5 V, 500 μ s) were applied to the gate 2,000 times, and the drain current was measured every 1 s with 1 V for 100 s. This figure is identical to Fig. 4c. **b** Newly measured retention characteristics for two states at various temperatures. Erase pulses (-3.3 V, 500 μ s) of 100 (red sphere) and 1,000 (blue sphere) were applied in the initial state, respectively, and read operations (1 V, 50 μ s) were performed every 1 s after the erase pulse train.

To reflect the above revisions, we replaced Fig. 3h (Fig. R10a) with Fig. R10b, and revised the sentences in the manuscript:

Page 12, line 225: “Retention characteristic of GIFET. Erase pulses (-3.3 V, 500 μ s) were applied to the gate to reach a certain conductance level, and read pulses (1 V, 50 μ s) were applied to the drain every 1 s subsequent to the erase pulse train. A 5.45% conductance value change was observed after 1,000 s.”

Page 14, line 267: “Figure 3h shows the data-holding ability of the GIFET. We observed a data loss of 5.45% (13.6 nS) of the updated conductance after 1,000 s. Also, several intermediate conductance levels were maintained without severe degradation.”

We also replaced Fig. 4c (Fig. R11a) with Fig. R11b to reflect new retention measurement data and, as the reviewer commented, we corrected the repeated character in Fig. 4d:

“

Fig. 4 Robustness of the GIFET to temperature variations. Linearity variation in various temperatures from 298 K to 393 K with **a** 1,000 potentiation pulses (4 V, 500 μ s) and **b** 1,000 depression pulses (-4 V, 500 μ s). **c** Measurement of retention characteristics of two states at various temperatures from 298 K to 393 K. Erase pulses (-3.3 V, 500 μ s) of 100 (red sphere) and 1,000 (blue sphere) were applied in the initial state, respectively, and read operations (1 V, 50 μ s) were performed every 1 s after the erase pulse train. **d** Comparison of endurance characteristics at different temperatures after 10^5 switching cycles (2×10^6 pulses). The whisker represents minimum and maximum values and the box range indicates drain current values in the 25% to 75% percentile.”

Page 15, line 280: “Measurement of retention characteristics of two conductance states at various temperatures from 298 K to 393 K. Erase pulses (-3.3 V, 500 μ s) of 100 (red sphere) and 1,000 (blue sphere) were applied in the initial state, respectively, and read operations (1 V, 50 μ s) were performed every 1 s after the erase pulse train.”

Page 16, line 293: “For verifying the reliability of multi-level conductance states in high temperatures, the retention characteristics were measured for two states according to temperature from 298 K to 393 K as shown in Fig. 4c. Erase pulses (-3.3 V, 500 μ s) of 100 (read sphere) and 1,000 (blue sphere) were applied in the initial state, respectively, and read

operations (1 V, 50 μ s) were performed every 1 s. It was confirmed that only the conductance level (drain current level) increased as the temperature increased and that the conductance states remained unchanged for 200 s. This indicates long-term plasticity properties, and the proposed GIFET can hold data for online training, even at high temperatures.”

Comment #8:

The endurance plot in Fig. 3(g) needs to be plotted with log x-axis to understand how the HRS/LRS change at different pulse numbers (low vs high), this is a standard methodology for showing logarithmic cycle numbers in endurance plots.

Response: We sincerely appreciate the reviewer for providing this generous comment. As shown in Fig. R12, we measured the higher endurance characteristics of GIFET than previous data and replaced Fig. 3g with Fig. R12 as mentioned above in comment 2. We also modified the x-axis of the endurance plot to a logarithmic scale. In addition, the endurance plots at various temperatures in Supplementary Fig. 16 were also modified to a logarithmic scale.

Fig. R12 Additional measured data on endurance characteristics. The endurance of GIFET over 2×10^8 pulses (2×10^5 switching cycles). Each switching cycle is composed of 500 write pulses with 5 V, 200 μ s and 500 erase pulses with -5 V, 200 μ s.”

Reviewer #3 (Remarks to the Author):

The authors have largely addressed the comments. I would like to recommend the publication of this revised manuscript.

Response: We would like to sincerely thank the reviewer for the valuable time the reviewer has spent reviewing our manuscript and providing insightful comments which greatly improve the quality of our work.

1. Yu, S. Neuro-inspired computing with emerging nonvolatile memories. *Proc. IEEE* **106**, 260–285 (2018).
2. Fuller, E. J. et al. Li-ion synaptic transistor for low power analog computing. *Adv. Mater.* **29**, 1604310 (2017).
3. Ambrogio, S. et al. Equivalent-accuracy accelerated neural-network training using analogue memory. *Nature* **558**, 60–67 (2018).
4. Zhang, W. et al. Neuro-inspired computing chips. *Nat. Electron.* **3**, 371–382 (2020).
5. Li, Y. et al. Capacitor-based cross-point array for analog neural network with record symmetry and linearity. In *2018 IEEE Symposium on VLSI Technology 25–26* (IEEE, 2018).
6. Chen, Z., Chen, X. & Gu, J. 15.3 a 65 nm 3T dynamic analog RAM-based computing-in-memory macro and CNN accelerator with retention enhancement, adaptive analog sparsity and 44TOPS/W system energy efficiency. In *2021 IEEE International Solid-State Circuits Conference (ISSCC) 240–242* (IEEE, 2021).
7. Ielmini, D. & Ambrogio, S. Emerging neuromorphic devices. *Nanotechnology* **31**, 92001 (2019).
8. Sun, X. & Yu, S. Impact of non-ideal characteristics of resistive synaptic devices on implementing convolutional neural networks. *IEEE J. Emerg. Sel. Top. Circuits Syst.* **9**, 570–579 (2019).
9. Yu, S., Jiang, H., Huang, S., Peng, X. & Lu, A. Compute-in-memory chips for deep learning: recent trends and prospects. *IEEE Circuits Syst. Mag.* **21**, 31–56 (2021).
10. Park, G. H. & Cho, W. J. Reliability of modified tunneling barriers for high performance nonvolatile charge trap flash memory application. *Appl. Phys. Lett.* **96**, 1–4 (2010).
11. Park, G. H., Jung, M. H., Kim, K. S., Chung, H. B. & Cho, W. J. Tunneling barrier engineered charge trap flash memory with ONO and NON tunneling dielectric layers. *Curr. Appl. Phys.* **10**, e13–e17 (2010).
12. Zhu, H. et al. Discrete charge states in nanowire flash memory with multiple Ta₂O₅ charge-trapping stacks. *Appl. Phys. Lett.* **104**, 1–6 (2014).
13. Lee, S. T. et al. Neuromorphic technology based on charge storage memory devices. In *2018 IEEE Symposium on VLSI Technology 169–170* (IEEE, 2018).
14. Zhao, M., Gao, B., Tang, J., Qian, H. & Wu, H. Reliability of analog resistive switching memory for neuromorphic computing. *Appl. Phys. Rev.* **7**, 011301 (2020).
15. Park, Y. J. et al. 3-D stacked synapse array based on charge-trap flash memory for implementation of deep neural networks. *IEEE Trans. Electron Devices* **66**, 420–427 (2019).
16. An, S., Lee, M., Park, S., Yang, H. & So, J. An ensemble of simple convolutional neural network models for MNIST digit recognition. Preprint at <https://arxiv.org/abs/2008.10400> (2020).
17. Byerly, A., Kalganova, T. & Dear, I. No routing needed between capsules.

Neurocomputing **463**, 545–553 (2021).

18. Hirata, D. & Takahashi, N. Ensemble learning in CNN augmented with fully connected subnetworks. Preprint at <https://arxiv.org/abs/2003.08562> (2020).
19. Chen, P.-Y., Peng, X. & Yu, S. NeuroSim+: An integrated device-to-algorithm framework for benchmarking synaptic devices and array architectures. *IEEE Int. Electron Devices Meeting (IEDM)* (IEEE, San Francisco, USA, 2017).
20. Yu, S. et al. Binary neural network with 16 Mb RRAM macro chip for classification and online training. In *2016 IEEE International Electron Devices Meeting (IEDM)* 16.2.1–16.2.4 (2016).
21. Kim, M. K. & Lee, J. S. Ferroelectric analog synaptic transistors. *Nano Lett.* **19**, 2044–2050 (2019).
22. Yu, R. et al. Electret-based organic synaptic transistor for neuromorphic computing. *ACS Appl. Mater. Interfaces* **12**, 15446–15455 (2020).
23. Wang, L. et al. Exploring ferroelectric switching in α -In₂Se₃ for neuromorphic computing. *Adv. Funct. Mater.* **30**, 1–9 (2020).
24. Chung, W., Si, M. & Peide, D. Y. First demonstration of Ge ferroelectric nanowire FET as synaptic device for online learning in neural network with high number of conductance state and g_{\max}/g_{\min} . In *2018 IEEE International Electron Devices Meeting (IEDM)* 15.2.1–15.2.4 (IEEE, 2018).
25. Chou, Y. C. et al. Neuro-inspired-in-memory computing using charge-trapping memtransistor on germanium as synaptic device. *IEEE Trans. Electron Devices* **67**, 3605–3609 (2020).

Reviewers' Comments:

Reviewer #2:

Remarks to the Author:

Summary: The authors have largely addressed all the comments and revised the manuscript accordingly. The revised version has stronger methods and is much more effective. The manuscript can be accepted with the following minor revisions:

1. It is better to put the HRS/LRS endurance cycles (10^5) rather than pulse number (10^8) in the benchmarking table.
2. How are the authors ensuring no interfacial layer in the structure shown in Fig. 2a during fabrication, that might be good to add.

Response Letter to Reviewers' Comments

We truly appreciate the reviewers for investing their valuable time and effort in reviewing our work and suggesting variety of insightful comments to improve the quality of our research. Considering reviewer #2's comments, we have made a point-by-point response letter and revised the main manuscript and supplementary materials to elaborate our research paper. We believe we have addressed reviewer #2's comments, and this would make the paper clearer and higher quality. Based on the two responses below, we have revised the table and appended some sentences in Supplementary Information. Our point-by-point response to reviewer #2's comments are as follows:

Reviewer #2 (Remarks to the Author):

Summary: The authors have largely addressed all the comments and revised the manuscript accordingly. The revised version has stronger methods and is much more effective. The manuscript can be accepted with the following minor revisions:

Response: We sincerely thank the reviewer for providing helpful comments on our revised manuscript. Based on the reviewer's valuable revisions, we were able to enhance and develop our manuscript to a much better quality until the end of the revision process. Our detailed responses to the reviewer's comments are provided below.

Comment #1

It is better to put the HRS/LRS endurance cycles (10^5) rather than pulse number (10^8) in the benchmarking table.

Response: We thank the reviewer for this insightful comment. As the reviewer mentioned, we agree to modify the value of the endurance characteristics in the benchmarking table to the HRS/LRS switching cycles. We reset the criterion from pulse number to switching cycles for a more equitable comparison because the endurance criterion by pulse numbers can tend to overstate the measured results. Therefore, we have revised the endurance characteristics of Table 1 on page 31, line 587 as below tables.

Before revision:

	[56]	[44]	[35]	[32]	[57]	[41]	[58]	This work
Endurance (Pulse Number)	N/A	360	12,800	$\sim 10^5$	N/A	4×10^3	N/A	2×10^8

After revision:

	[56]	[44]	[35]	[32]	[57]	[41]	[58]	This work
Endurance (HRS/LRS switching cycles)	N/A	N/A	$> 10^5$	> 400	N/A	> 40	N/A	$> 2 \times 10^5$

Comment #2

How are the authors ensuring no interfacial layer in the structure shown in Fig. 2a during fabrication, that might be good to add.

Response: We would like to thank the reviewer for providing this valuable comment. As the reviewer was concerned, it is clearer to mention the sample preparation process. In Fig 2a, before WO_x deposition, we applied a chemical solution to make a clean surface. The CSL (WO_x) layer was deposited right after cleaning the sample to minimize the effect of native oxide effect.

We added the sentence in the Supplementary Fig. 3 section as shown below.

“A chemical solution was applied to ensure a clean surface prior to WO_x deposition. A CSL(WO_x) layer was deposited immediately after sample cleaning to minimize the effect of native oxide.”